

# Data assimilation as a deep learning tool to infer ODE representations of dynamical models

Marc Bocquet[1], Julien Brajard[2,3], Alberto Carrassi[3,4], and Laurent Bertino[3]

[1]CEREA, joint laboratory École des Ponts ParisTech and EDF R&D, Université Paris-Est, Champs-sur-Marne, France
[2]Sorbonne University, CNRS-IRD-MNHN, LOCEAN, Paris, France
[3]Nansen Environmental and Remote Sensing Center, Bergen, Norway
[4]Geophysical Institute, University of Bergen, Norway

**Correspondence:** M. Bocquet (marc.bocquet@enpc.fr)

**Abstract.** Recent progress in machine learning has shown how to forecast and, to some extent, learn the dynamics of a model from its output, resorting in particular to neural networks and deep learning techniques. We will show how the same goal can be directly achieved using data assimilation techniques without leveraging on machine learning software libraries, with a view to high-dimensional models. The dynamics of a model are learned from its observation and an ordinary differential equation

(ODE) representation of this model is inferred using a recursive nonlinear regression. Because the method is embedded in a Bayesian data assimilation framework, it can learn from partial and noisy observations of a state trajectory of the physical model. Moreover, a space-wise local representation of the ODE system is introduced and is key to cope with high-dimensional models.

It has recently been suggested that neural network architectures could be interpreted as dynamical systems. Reciprocally, we

show that our ODE representations are reminiscent of deep learning architectures. Furthermore, numerical analysis considerations on stability shed light on the assets and limitations of the method.

The method is illustrated on several chaotic discrete and continuous models of various dimensions, with or without noisy observations, with the goal to identify or improve the model dynamics, build a surrogate or reduced model, or produce forecasts from mere observations of the physical model.

*Copyright statement.*

## 1  Introduction

### 1.1  Data assimilation and model error

Data assimilation aims at estimating the state of a physical system from observations and a numerical dynamical model of that system. It has been successfully applied to numerical weather and ocean prediction, where it often consisted in estimating

the initial conditions for the state trajectory of chaotic geofluids (Kalnay, 2002; Asch et al., 2016; Carrassi et al., 2018).



This objective is mostly impeded by the deficiencies of the numerical model (discretrisation, approximate physical schemes, unresolved scales and their uncertain parametrisations, e.g., Harlim, 2017) and the difficulty to match numerical representations of the system with the observations (representation error, Janjić et al., 2018). As a result, the quality of numerical weather predictions based on a methodologically sound data assimilation method crucially depends both on the sensitivity to initial

condition due to the chaotic unstable dynamics and on model error (Magnusson and Källén, 2013).

Model errors can take many forms and accounting for them depends on the chosen data assimilation scheme. A first class of solutions rely on parametrising model error by, for instance, transforming the problem into a physical parameter estimation problem (e.g., Bocquet, 2012; Aster et al., 2013). Other solutions are based on a weakly parametrised form of model error, for instance when it is assumed to be additive noise. Such techniques have been developed for variational data assimilation (e.g.,

Trémolet, 2006; Carrassi and Vannitsem, 2010), for ensemble Kalman filters and smoothers (e.g., Ruiz et al., 2013; Raanes et al., 2015) and ensemble-variational assimilation (Amezcua et al., 2017; Sakov et al., 2018). In the weakly parametrised form, these techniques should be completed by an estimation of the model error statistics (e.g., Pulido et al., 2018; Tandeo et al., 2018). Moreover, model error's impact can be mitigated by multiplicative and additive inflation (e.g., Whitaker and Hamill, 2012; Grudzien et al., 2018; Raanes et al., 2019) and by physically-driven stochastic perturbations of model simulations in

ensemble approaches (e.g., Buizza et al., 1999), or by stochastic subgrid parametrisations (e.g., Resseguier et al., 2017). This account is very far from exhaustive as this is a vast, multiform and very active subject of research.

These approaches essentially seek to correct, calibrate or improve an existing model using observations. Hence, they all primarily make use of data assimilation techniques.

### 1.2   Data-driven forecast of a physical system

Another approach is to renounce to physically-based numerical models of the phenomenon of interest and instead use observations only of that system. Given the huge required datasets, this may seem a far-reaching goal for operational weather and ocean forecasting systems, but recent progress in data-driven methods and convincing applications to geophysical problems of small to intermediate complexity are strong incentives to investigate this bolder approach. Eventually, the perspective of putting numerical models away has a strong practical appeal, even though such perspective may generate intense debates.

For instance, forecasting of a physical system can be done by looking up at past situations and patterns using the techniques of analogues, which can be combined with present observations using data assimilation (Lguensat et al., 2017). It can rely on a representation of the physical system based on diffusion maps that look for a spectral representation of the dataset (see chapter 6 of Harlim, 2018). An original data-driven stochastic modelling approach has been developed by Kondrashov et al. (2015). The method, recently extended to deal with multi-scale datasets (Kondrashov and Chrekroun, 2017), has been applied

to successfully estimate reduced models of geophysical phenomena (see e.g., Kondrashov et al., 2018, and references therein). A fourth route relies on neural networks and deep learning to represent the hidden model and make forecasts from this representation. Examples of such approach applied to the forecasting of low-order chaotic geophysical models are: Park and Zhu (1994) who use a bilinear recurrent neural network and applied it on the three-variable Lorenz model (Lorenz, 1963, hereafter L63), Pathak et al. (2017, 2018) who use reservoir network techniques on the L63 model and on the Kuramoto-Sivashinski




model (Kuramoto and Tsuzuki, 1976; Sivashinsky, 1977, hereafter KS); Dueben and Bauer (2018) who use a neural network on a low-order Lorenz three-scale model, and on coarse 2D geopotential height maps at 500 hPa. The last two contributions have to resort to local reservoir networks or convolutional layers, respectively, to account for the dimensionality of the models. However, all these representations are not explicit and the neural network becomes a surrogate for the hidden model. This

marks a key distinction with respect to our approach, as will be clarified later.

### 1.3   Learning the dynamics of a model from its output

Data-driven techniques that seek to represent the model in a more explicit manner, and therefore with a greater interpretability, may use specific classes of nonlinear regression as advocated by Paduart et al. (2010); Brunton et al. (2016). With a view to forecast dynamical systems, it is possible to design neural networks in order to reflect the iterative scheme of a Runge-Kutta

(RK) integration scheme. Wang and Lin (1998) proposed and achieved such a goal using classical activation functions, which may however blur the interpretation of the underlying dynamics. Fablet et al. (2018) went further and used a bilinear residual neural network structured to mimic a fourth-order Runge-Kutta scheme (RK4) and noise-free data. Using the Keras tool with the TensorFlow backend, their approach proved to be a very effective tool for the L63 model and to a lesser extent to the 40-variable Lorenz model (Lorenz and Emanuel, 1998, hereafter L96). In particular, they retrieved the parameters of the L63

equations to a high precision. Long et al. (2018) sought the operators of the partial differential equations (PDEs) of a physical system by identifying differentiations with convolution operators of a feed-forward neural network. They successfully applied their method to advection-diffusion problems. Note that, as opposed to our proposal, none of the aforementioned techniques is embedded in a Bayesian framework, making them less suitable to work with noisy and partial data.

### 1.4   Goal and outline

From this point on, the physical system under scrutiny will be called the *reference* model. It will be assumed to be known only from observations. We follow a data-driven approach inspired by the work of Paduart et al. (2010); Fablet et al. (2018) in the sense that we will consider an observed physical reference model, which might be generated by a hidden mathematical model. This work is focused on either one or a combination of the following goals: (i) to build a *surrogate* model for the dynamics, (ii) to produce forecasts that emulate those of the reference model, and (iii) to identify the underlying dynamics of the reference

model given by a mathematical model. The reference model could be totally unknown or only partially specified.

To achieve these goals, we introduce a surrogate model defined by a set of ordinary differential equations (ODEs):

$$\frac{d\mathbf{x}}{dt} = \phi(\mathbf{x}), \tag{1}$$

where $\mathbf{x} \in \mathbb{R}^{N_x}$ is the state vector, and $\mathbf{x} \mapsto \phi(\mathbf{x})$ is a vector field that we shall call *flow rate*. For the sake of simplicity, the dynamics in this paper are supposed to be autonomous, i.e. do not explicitly depend on time. Our technique seeks a fit $\phi$ given

observations of the reference model. This is a rather general representation since, for instance, PDEs can be discretised into ODEs. We will restrict ourselves to the case where $\phi$ is at most quadratic in $\{x_n\}_{0 \le n < N_x}$. The numerical integration of Eq. (1)



could be based on any RK scheme as an integration, but should additionally rely on the composition of such integration steps. As a result, quite general resolvents of Eq. (1) can be built.

Importantly, we will not require any machine learning software tool since the adjoint of the model resolvent can be derived without too much of an effort. As opposed to the contributions mentioned in the previous subsections, we embed the technique

in a data assimilation framework. From a data assimilation standpoint, the technique can be seen as meant to deal with model error (with or without some prior on the model) and it naturally allows to use partial and noisy observations. We will also build representations of the dynamics that are either invariant by spatial translation (homogeneous) and/or local (i.e. the flow rate of a variable $x_n$ only depends on neighbouring variables whose perimeter is defined by a stencil). These properties make our technique scalable and thus potentially applicable to high-dimensional systems.

In Sect. 2, we present model identification as a Bayesian data assimilation problem. We first choose an ODE representation of the dynamics, introduce a nonlinear regressor basis, and define integration schemes we will work with. We describe the local and homogeneous representations as physically-based simplifications of the most general case, and we build the adjoint of these representations. We then introduce the Bayesian problem and the resulting cost function used for joint supervised learning of the optimal representation and estimation of the state trajectory. The latter is the standard goal of data assimilation

while the former is that of machine learning. Our approach blend them together using the formalism of data assimilation.

In Sect. 3, we discuss several theoretical issues: convergence of the learning step, the connection with numerical analysis of integration schemes, the connection with deep learning architectures, and, finally, the pros and cons of our approach.

In Sect. 4, we illustrate the method with several low-order chaotic models (L63, L96, KS and a two-scale Lorenz model) of various sizes, from a a perfectly identifiable model, i.e. where the model used to generate the dataset can be retrieved exactly,

to the design of a reduced-order model where the model used to generate the dataset cannot be retrieved exactly, using full or partial, noiseless or noisy observations. Conclusions are given in Sect. 5.

## 2    Model identification as a data assimilation problem

### 2.1    Ordinary differential equation representation

Our surrogate model is chosen to be represented by an ODE system as described by Eq. (1). We assume that the flow rate can

be written

$$\phi_{\mathbf{A}}(\mathbf{x}) = \mathbf{A}\mathbf{r}(\mathbf{x}), \qquad (2)$$

where $\mathbf{A} \in \mathbb{R}^{N_x \times N_p}$ is a matrix of real coefficients to be estimated and $\mathbf{r} : \mathbb{R}^{N_x} \mapsto \mathbb{R}^{N_p}$ is a map that defines regressor functions of $\mathbf{x}$. $\mathbb{R}^{N_p}$ is the latent space of the regressors in which the flow rate is linear.

In the absence of any peculiar symmetry, we choose this map to list all the monomials up to second-order built on $\mathbf{x}$, i.e the

constant, linear and bilinear monomials. Let us call $\mathcal{D} = \{0, 1, \dots, N_x - 1\}$ the set of all variable indices, and $\mathcal{P}$ the set of all



pairs of variable indices. We introduce the augmented state vector

$$\tilde{\mathbf{x}} = \begin{bmatrix} \mathbf{x} \\ 1 \end{bmatrix} \in \mathbb{R}^{N_x+1}, \tag{3}$$

extend $\mathcal{D}$ to $\widetilde{\mathcal{D}} = \mathcal{D} \cup \{N_x\}$, and define $\widetilde{\mathcal{P}}$ as the distinct pairs of variable indices in $\widetilde{\mathcal{D}}$.

As a result, the regressors are compactly defined by

$$\mathbf{r}(\mathbf{x}) = \left[ \{\tilde{x}_n \tilde{x}_m\}_{(n,m) \in \widetilde{\mathcal{P}}} \right] \tag{4}$$

where the scalars in the bracket are the entries of the vector $\mathbf{r}(\mathbf{x})$. We count

$$N_p = \binom{N_x+1}{2} = \frac{1}{2}(N_x+1)(N_x+2) \tag{5}$$

regressors, i.e. the cardinal of $\widetilde{\mathcal{P}}$. Higher-order regressors could be included, or of different functional form as in Brunton et al. (2016). However, it is important to keep in mind that we do not seek an expansion of the resolvent of the reference model but of the flow rate $\boldsymbol{\phi_A}$. Furthermore, higher-order products of the state variables are later generated by integration schemes and their composition. It is worth mentioning that nonlinear regressions are not widespread in geophysical data assimilation. We are nonetheless aware of at least one noticeable exception that extends traditional Gaussian-based methods (Hodyss, 2011, 2012).

## 2.2 Local and homogeneous representations

At least two useful simplifications for the ODEs could be exploited if the state $\mathbf{x}$ is assumed to be the discretisation of a spatial field.

### 2.2.1 Locality

First, we use a locality assumption based on the physical locality of the system: all multivariate monomials in the ODEs have variables $x_n$ that belong to a stencil, i.e. a local arrangement of grid points around a given node. This can significantly reduce the number of bilinear monomials in $\mathbf{r}(\mathbf{x})$. We assume that $s_n$ is the stencil around node $n$, the pattern being the same for all nodes. For the node $N_x$ corresponding to the extra variable $\tilde{x}_{N_x} = 1$, we assume that its stencil consists of all the $N_x + 1$ nodes. We can then define $\widetilde{\mathcal{P}}_s \subset \widetilde{\mathcal{P}}$ as the sub-set of all pairs $(n, m)$ of variables for which $m \in s_n$. The set of required monomials can therefore be reduced to

$$\mathbf{r}(\mathbf{x}) = \left[ \{\tilde{x}_n \tilde{x}_m\}_{(n,m) \in \widetilde{\mathcal{P}}_s} \right]. \tag{6}$$

Under these conditions, $\mathbf{A}$ becomes sparse. Indeed, for each node $n$, we assume that $\dot{x}_n$, the time derivative of $x_n$, is impacted only by linear terms $x_m$ such that $m \in s_n$ and quadratic terms $x_m x_l$ such that $m \in s_n$, $l \in s_n$ and $m \in s_l$. However, to keep a dense matrix, we choose to compactly redefine and shrink $\mathbf{A}$ by eliminating all a priori zero entries due to the locality



assumption. The number of columns of $\mathbf{A}$ is then significantly reduced from $N_p$ to $N_a$. As a consequence of this redefinition of $\mathbf{A}$, the matrix multiplication inbetween $\mathbf{A}$ and $\mathbf{r}(\mathbf{x})$ must be changed accordingly. Nonetheless, the operation that assigns coefficients in $\mathbf{A}$ to the monomials in $\mathbf{r}(\mathbf{x})$ remain linear and we write it as

$$\phi_{\mathbf{A}}(\mathbf{x}) = \mathbf{A} \bullet \mathbf{r}(\mathbf{x}). \tag{7}$$

Let us take the case of a one-dimensional state space as used in Sect. 4. The domain is supposed to be periodic and the nodes are indexed by $0 \leq n < N_x$. The node of index $N_x$ is associated to the extra $\{1\}$. For $0 \leq n < N_x$, the stencil $s_n$ is defined as the set of $2L+1$ nodes of index $n-L, n-L+1, \ldots n+L-1, n+L$, plus the extra node of index $N_x$. The stencil $s_{N_x}$ consists of all the nodes, i.e. $\mathcal{D}$. We assume $2L+1 \leq N_x$. In that case $\mathbf{r}(\mathbf{x})$ as defined by Eq. (6) has $N_p = 1 + N_x(2+L)$ monomials.

The row $[\mathbf{A}]_n$ of the dense $\mathbf{A}$ contains the following coefficients for each $0 \leq n < N_x$. First there are $2L+2$ regres-
sors built with $\{1\}$ (the constant and linear regressors). Second, we consider the square monomials $x_m^2$ with $m \in s_n$, i.e. $\left\{ x_m^2 \right\}_{n-L \leq m \leq n+L}$ whose number is $2L+1$. Then we consider those separated by one step: $\{x_m x_{m+1}\}_{n-L \leq m \leq n+L-1}$ whose number is $2L$, followed by those separated by two steps whose number is $2L-1$, and so on until a separation of $L$ is reached. Quadratic monomials of greater separation are discarded since they do not belong to a common stencil as per the above definition reflecting the locality assumption. Hence there is a total of $N_a = \sum_{l=L+1}^{2L+2} l = \frac{3}{2}(L+1)(L+2)$ coefficients per grid-cell.
Note that this locality assumption is hardly restrictive. Indeed, owing to the absence of long-range instantaneous interactions (which are precluded in geophysical fluids), farther distance correlations between state variables can be generated by small stencils in the definition of $\phi_{\mathbf{A}}$ through time integrations.

### 2.2.2 Homogeneity

Furthermore, a symmetry hypothesis could optionally be used by assuming translational invariance of the ODEs, called the
*homogeneity* assumption in the following. Because our control parameters, i.e. the coefficients of $\mathbf{A}$ parametrise the flow rate, the symmetry simply translates into the rows $[\mathbf{A}]_n$ of the dense $\mathbf{A}$ being the same for all $n$. Hence $\mathbf{A}$ simply becomes a vector in $\mathbb{R}^{N_a}$.

Note that, while both constraints, locality and homogeneity, apply to the ODEs, they do not apply to the states per se. For instance ODEs for discretised homogeneous two-dimensional turbulence satisfy both constraints and yet generate hardly trivial
flows.

In appendix A, we show in the one-dimensional case how to compute the reduced form of the product between $\mathbf{A}$ and $\mathbf{r}(\mathbf{x})$, in the presence of both locality and homogeneity assumptions. This type of technical parametrisation is required for a parsimonious representation of the control variables, i.e. the coefficients of $\mathbf{A}$, and is key for a successful implementation in high-dimensional models.

### 30   2.3   Integration scheme and cycling

The reference model will be observed at time steps $t_k$, indexed by integer $0 \leq k \leq K$. Hence, we need to be able to express the resolvent of the surrogate model from $t_k$ to $t_{k+1}$. We assume that $t_{k+1} - t_k$ is a multiple of the integration time step of the



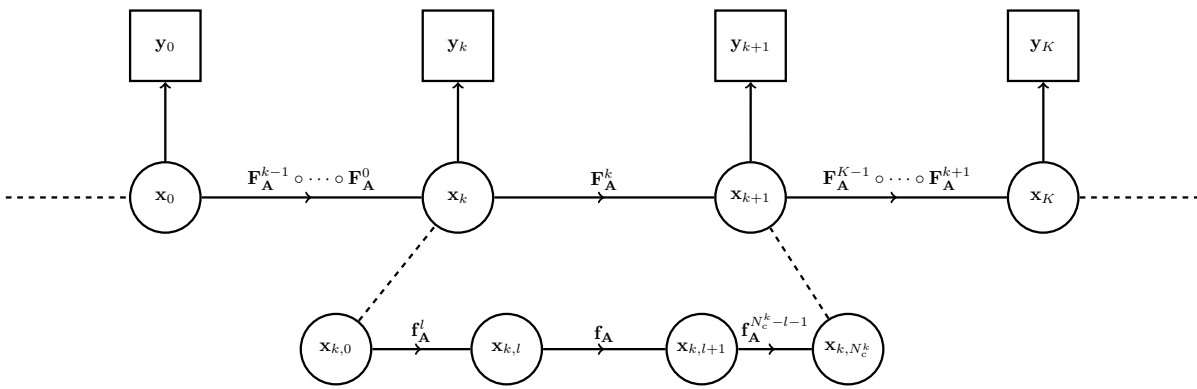

**Figure 1.** Representation of the data assimilation system as a hidden Markov chain model, and of the model resolvents $\mathbf{F}_\mathbf{A}^k$ and $\mathbf{f}_\mathbf{A}$.

surrogate model: $t_{k+1} - t_k = N_c^k h$, where $h$ is the integration time step and $N_c^k$ is the number of integrations. The time steps $t_{k+1} - t_k$ can be uneven, which is reflected in the dependence of $N_c^k$ on $k$. Hence, the resolvent of the surrogate model from $t_k$ to $t_{k+1}$ can be written as

$$\mathbf{x}_{k+1} = \mathbf{F}_\mathbf{A}^k(\mathbf{x}_k) \quad \text{where} \quad \mathbf{F}_\mathbf{A}^k \equiv \mathbf{f}_\mathbf{A}^{N_c^k} \equiv \underbrace{\mathbf{f}_\mathbf{A} \circ \ldots \circ \mathbf{f}_\mathbf{A}}_{N_c^k \text{ times}}, \tag{8}$$

meant to be the integration of Eq. (1) from $t_k$ to $t_{k+1}$ using the representation Eq. (2).

We define intermediate state vectors inbetween $[t_k, t_{k+1}]$: $\mathbf{x}_{k,l}$ is the state vector defined at time $t_k + (t_{k+1} - t_k)l/N_c^k$ for $0 \le l \le N_c^k$, as the result of $l$ compositions of $\mathbf{f}_\mathbf{A}$ on $\mathbf{x}_k$: $\mathbf{x}_{k,l} = \mathbf{f}_\mathbf{A}^l(\mathbf{x}_k)$. Figure 1 is a representation of the composition of the integration steps, along with the state vectors $\mathbf{x}_k$ and $\mathbf{x}_{k,l}$.

The operator $\mathbf{f}_\mathbf{A}$ is meant to be an explicit numerical integration scheme. We shall consider a Runge-Kutta (RK) scheme in
the following applied to $\mathbf{x} \equiv \mathbf{x}_{k,l}$. It has $N_{\mathrm{RK}}$ steps. This number of steps coincides with the accuracy of the scheme for those that we consider in the following: first-order for the Euler scheme, second-order for RK2, and fourth-order for RK4. Provided the dynamics are autonomous, a general RK scheme reads

$$\mathbf{f}_\mathbf{A}(\mathbf{x}) = \mathbf{x} + h \sum_{i=0}^{N_{\mathrm{RK}}-1} \beta_i \mathbf{k}_i, \tag{9a}$$

$$\mathbf{k}_i = \phi_\mathbf{A}\left(\mathbf{x} + h \sum_{j=0}^{i-1} \alpha_{i,j} \mathbf{k}_j\right), \tag{9b}$$

where the coefficients $\beta_i$ and $\alpha_{i,j}$ entirely specify the scheme and $h = (t_{k+1} - t_k)/N_c^k$. Note that $\alpha_{i,j}$ are zero for $j \ge i$, so that Eq. (9b) can be computed iteratively from $\mathbf{k}_0$ to $\mathbf{k}_{N_{\mathrm{RK}}-1}$ followed by the sum Eq. (9a) to get $\mathbf{f}_\mathbf{A}(\mathbf{x})$.

It will be useful in the following to consider the variation of the $\mathbf{k}_i$ with respect to either $\mathbf{A}$ or $\mathbf{x}$:

$$\delta \mathbf{k}_i = \delta \phi_{\mathbf{A},i} + h \sum_{j=0}^{i-1} \alpha_{i,j} \left(\nabla_{\mathbf{x}_i} \phi_{\mathbf{A},i}\right) \delta \mathbf{k}_j, \tag{10}$$




where $\phi_{\mathbf{A},i}$ is $\phi_{\mathbf{A}}$ evaluated at $\mathbf{x}_i^{\mathrm{RK}} \equiv \mathbf{x} + h\sum_{j=0}^{i-1}\alpha_{i,j}\mathbf{k}_j$, $\nabla_{\mathbf{x}_i}\phi_{\mathbf{A},i}$ is the tangent linear with respect to $\mathbf{x}$ of $\phi_{\mathbf{A}}$ evaluated at $\mathbf{x}_i^{\mathrm{RK}}$. Equation (10) can be written compactly in the form

$$\mathbf{G}\delta\boldsymbol{\kappa} = \delta\boldsymbol{\varphi}, \tag{11}$$

where $\mathbf{G}$ is the matrix of size $(N_{\mathrm{RK}}N_x) \times (N_{\mathrm{RK}}N_x)$ defined by its $N_x \times N_x$ blocks $[\mathbf{G}]_{i,j} = \mathbf{I}_x - h\alpha_{i,j}\nabla_{\mathbf{x}_i}\phi_{\mathbf{A},i}$, $\boldsymbol{\kappa}$ is the

vector of size $N_{\mathrm{RK}}N_x$ which results from the stacking of the $\mathbf{k}_i \in \mathbb{R}^{N_x}$ for $0 \le i < N_{\mathrm{RK}}$, $\boldsymbol{\varphi}$ is the vector of size $N_{\mathrm{RK}}N_x$ which results from the stacking of the $\phi_{\mathbf{A},i}$ for $0 \le i < N_{\mathrm{RK}}$, and $\mathbf{I}_x \in \mathbb{R}^{N_x}$ is the identity matrix. The important point is that $\mathbf{G}$ is a lower triangular matrix and describes an iterative construction of the $\mathbf{k}_i$. Moreover, the diagonal entries of $\mathbf{G}$ are 1 by construction so that $\mathbf{G}$ is invertible and

$$\delta\boldsymbol{\kappa} = \mathbf{G}^{-1}\delta\boldsymbol{\varphi}. \tag{12}$$

This will be useful to evaluate the variations of $\mathbf{f}_{\mathbf{A}}(\mathbf{x})$ via Eq. (9a).

In the following, $h$ will be absorbed in the definition of $\mathbf{A}$ and hence $\phi_{\mathbf{A}}$, so that we can take $h = 1$ without loss of generality.

## 2.4    Bayesian analysis

We consider a sequence of observation vectors $\mathbf{y}_k \in \mathbb{R}^{N_y^k}$ of the physical system at $t_k$ indexed by $0 \le k \le K$. The system state is observed through

$$\mathbf{y}_k = \mathbf{H}_k(\mathbf{x}_k) + \boldsymbol{\epsilon}_k, \tag{13}$$

where $\mathbf{H}_k$ is the observation operator at time $t_k$. The observation error $\boldsymbol{\epsilon}_k$ will be assumed Gaussian of zero mean and covariance matrix $\mathbf{R}_k$, It is also assumed to be white in time. Since the flow rate $\phi_{\mathbf{A}}$ is given by the approximation Eq. (2), so is the resolvent $\mathbf{F}_{\mathbf{A}}$ of the surrogate model. Hence, we generalise Eq. (8) to

$$\mathbf{x}_{k+1} = \mathbf{F}_{\mathbf{A}}^k(\mathbf{x}_k) + \boldsymbol{\eta}_k, \tag{14}$$

where $\boldsymbol{\eta}_k$ are unbiased Gaussian errors of covariance matrices $\mathbf{Q}_k$, supposed to be white in time. Note that, in all generality, the state space of the surrogate model does not have to match that of the reference model. We will nonetheless take them to coincide here only for simplicity.

With the goal to identify a model, or build a surrogate of the reference one, we are interested in estimating the probability density function (pdf) $p(\mathbf{A}|\mathbf{y}_{0:K})$ where $\mathbf{y}_{0:K}$ stands for all observations in the window $[t_0, t_K]$. To obtain a tractable expression

for this conditional likelihood, we need to marginalise over the state variables $\mathbf{x}_{0:K}$ within the window:

$$p(\mathbf{A}|\mathbf{y}_{0:K}) = \int d\mathbf{x}_{0:K}\, p(\mathbf{A}, \mathbf{x}_{0:K}|\mathbf{y}_{0:K}). \tag{15}$$





An approximate maximum a posteriori for $\mathbf{A}$ could be obtained by using the Laplace approximation of this integral, and hence getting the maximum of

$$
\begin{aligned}
p(\mathbf{A}, \mathbf{x}_{0:K}|\mathbf{y}_{0:K}) &= \frac{p(\mathbf{A}, \mathbf{x}_{0:K}, \mathbf{y}_{0:K})}{p(\mathbf{y}_{0:K})} \\
&= \frac{p(\mathbf{x}_{0:K}, \mathbf{y}_{0:K}|\mathbf{A})p(\mathbf{A})}{p(\mathbf{y}_{0:K})} \\
&= \frac{p(\mathbf{y}_{0:K}|\mathbf{x}_{0:K}, \mathbf{A})p(\mathbf{x}_{0:K}|\mathbf{A})p(\mathbf{A})}{p(\mathbf{y}_{0:K})}.
\end{aligned}
\tag{16}
$$

The cost function associated to this joint pdf is by definition $\mathcal{J}(\mathbf{A}, \mathbf{x}_{0:K}) = -\ln(p(\mathbf{A}, \mathbf{x}_{0:K}|\mathbf{y}_{0:K}))$. Under the Markovian assumption of Eq. (14) and the Gaussian form of both model and observational errors, the cost function reads

$$
\begin{aligned}
\mathcal{J}(\mathbf{A}, \mathbf{x}_{0:K}) =& \frac{1}{2}\sum_{k=0}^{K} \|\mathbf{y}_k - \mathbf{H}_k(\mathbf{x}_k)\|_{\mathbf{R}_k^{-1}}^2 \\
&+ \frac{1}{2}\sum_{k=1}^{K} \left\|\mathbf{x}_k - \mathbf{F}_{\mathbf{A}}^{k-1}(\mathbf{x}_{k-1})\right\|_{\mathbf{Q}_k^{-1}}^2 \\
&- \ln p(\mathbf{x}_0|\mathbf{A}) - \ln p(\mathbf{A}),
\end{aligned}
\tag{17}
$$

up to a constant depending on $\mathbf{Q}_{1:K}$ and $\mathbf{R}_{0:K}$ only. The vector norm $\|\mathbf{z}\|_{\mathbf{P}}$ is defined as $\sqrt{\mathbf{z}^{\mathbf{T}}\mathbf{P}\mathbf{z}}$. This is the cost function of a weak constraint 4D-Var (see Sect. 2.4.3.2 of Asch et al., 2016) with $\mathbf{A}$ and $\mathbf{x}_{0:K}$ as control variables.

In the case where the reference model is fully observed, i.e. $\mathbf{H}_k \equiv \mathbf{I}_x$ and in the absence of observation noise, i.e. $\mathbf{R}_k \equiv \mathbf{0}$, we have $\mathbf{x}_{0:K} \equiv \mathbf{y}_{0:K}$ and the cost function simplifies to

$$
\begin{aligned}
\mathcal{J}(\mathbf{A}) =& \frac{1}{2}\sum_{k=1}^{K} \left\|\mathbf{y}_k - \mathbf{F}_{\mathbf{A}}^{k-1}(\mathbf{y}_{k-1})\right\|_{\mathbf{Q}_k^{-1}}^2 \\
&- \ln p(\mathbf{y}_0|\mathbf{A}) - \ln p(\mathbf{A}),
\end{aligned}
\tag{18}
$$

where $\mathbf{y}_{0:K}$ is the fully and perfectly observed state trajectory of the reference model. This is similar to a traditional least-square function used in machine and deep learning regression. Reciprocally, when the aforementioned hypotheses of noiseless and complete observations do not hold, Eq. (17) can be seen as a natural data assimilation extension of Eq. (18). Note that Eq. (18) only depends on the sequence $\mathbf{Q}_{1:K}$. If, in addition, the dependence on the prior $p(\mathbf{y}_0, \mathbf{A})$ is neglected in Eq. (18), then the maximum a posteriori should not depend on a global rescaling of $\mathbf{Q}_{1:K}$. This connection between machine learning and data assimilation cost functions had been previously put forward by Abarbanel et al. (2018) although in a different form.

The data assimilation system is represented in Fig. 1 as a hidden Markov chain model. This Bayesian view highlights the choice that must be made for $\mathbf{R}_{0:K}$ and/or $\mathbf{Q}_{1:K}$ and provides an interpretation in terms of errors. Furthermore, one could implement an objective estimation of these error statistics as in Pulido et al. (2018).

The reader not interested by the following technicalities may skip the next subsection 2.5 and jump to Sect. 3. Nonetheless, they are critical to the success of the method.




## 2.5 Gradients and adjoint of the representation

To efficiently minimise the cost function Eq. (17) with gradient-based optimisation tool, we need to derive the gradient of Eq. (17) with respect to both $\mathbf{A}$ and $\mathbf{x}_{0:K}$. As for $\mathbf{x}_{0:K}$, we have:

$$\nabla_{\mathbf{x}_0}\mathcal{J} = -\left(\nabla_{\mathbf{x}_0}\mathbf{H}_0\right)^{\mathrm{T}}\mathbf{R}_0^{-1}\boldsymbol{\delta}_0 - \left(\nabla_{\mathbf{x}_0}\mathbf{F}_{\mathbf{A}}^0\right)^{\mathrm{T}}\mathbf{Q}_1^{-1}\boldsymbol{\Delta}_1$$

$$\qquad\qquad - \nabla_{\mathbf{x}_0}\ln p(\mathbf{x}_0|\mathbf{A}), \tag{19a}$$

$$\nabla_{\mathbf{x}_k}\mathcal{J} = -\left(\nabla_{\mathbf{x}_k}\mathbf{H}_k\right)^{\mathrm{T}}\mathbf{R}_k^{-1}\boldsymbol{\delta}_k - \left(\nabla_{\mathbf{x}_k}\mathbf{F}_{\mathbf{A}}^k\right)^{\mathrm{T}}\mathbf{Q}_{k+1}^{-1}\boldsymbol{\Delta}_{k+1}$$

$$\qquad\qquad + \mathbf{Q}_k^{-1}\boldsymbol{\Delta}_k, \quad \text{for} \quad 1 \le k \le K-1, \tag{19b}$$

$$\nabla_{\mathbf{x}_K}\mathcal{J} = -\left(\nabla_{\mathbf{x}_K}\mathbf{H}_K\right)^{\mathrm{T}}\mathbf{R}_K^{-1}\boldsymbol{\delta}_K + \mathbf{Q}_K^{-1}\boldsymbol{\Delta}_K, \tag{19c}$$

where $\boldsymbol{\delta}_k = \mathbf{y}_k - \mathbf{H}_k(\mathbf{x}_k)$ for $0 \le k \le K$ and $\boldsymbol{\Delta}_k = \mathbf{x}_k - \mathbf{F}_{\mathbf{A}}^{k-1}(\mathbf{x}_{k-1})$ for $1 \le k \le K$; $\nabla_{\mathbf{x}_k}\mathbf{F}_{\mathbf{A}}^k$ is the tangent linear of the resolvent $\mathbf{F}_{\mathbf{A}}^k$ computed at $\mathbf{x}_k$ for $0 \le k < K$; $\nabla_{\mathbf{x}_k}\mathbf{H}_k$ is the tangent linear of the observation operator $\mathbf{H}_k$ computed at $\mathbf{x}_k$ for $0 \le k \le K$. As for $\mathbf{A}$, we have:

$$\nabla_{\mathbf{A}}\mathcal{J} = -\sum_{k=1}^{K}\boldsymbol{\delta}_k^{\mathrm{T}}\mathbf{Q}_k^{-1}\nabla_{\mathbf{A}}\mathbf{F}_{\mathbf{A}}^{k-1}(\mathbf{x}_{k-1}) - \nabla_{\mathbf{A}}\ln p(\mathbf{A}), \tag{20}$$

assuming $\mathbf{x}_0$ is independent from $\mathbf{A}$. Hence, a key technical aspect of the problem is to compute the tangent linear and adjoint operators required by these gradients. In the following, we assume that the adjoint of the tangent linear operators of the observation operators $(\nabla_{\mathbf{x}_k}\mathbf{H}_k)^{\mathrm{T}}$ are known, for instance if the latter are linear as in Sect. 4.

### 2.5.1 Integration step

We first consider the situation when the observation interval corresponds to one integration time step of the surrogate model, i.e. $N_{\mathrm{c}}^k = 1$: $\mathbf{x}' = \mathbf{F}_{\mathbf{A}}(\mathbf{x}) = \mathbf{f}_{\mathbf{A}}(\mathbf{x})$ with $\mathbf{x} \equiv \mathbf{x}_k$. As a result, the time index $k$ can be omitted here. We will later consider the composition of several integration schemes ($N_{\mathrm{c}} \ge 2$). Equation (9a) is written again but as

$$\mathbf{f}_{\mathbf{A}}(\mathbf{x}) = \mathbf{x} + \mathbf{b}\boldsymbol{\kappa}, \tag{21}$$

where $\mathbf{b} = \boldsymbol{\beta} \otimes \mathbf{I}_x$ is the matrix of size $N_x \times (N_{\mathrm{RK}}N_x)$ tensor product of the vector $\boldsymbol{\beta}$ defined by $\boldsymbol{\beta}^{\mathrm{T}} = (\beta_0, \ldots, \beta_{N_{\mathrm{RK}}-1})$, i.e. the coefficients of the RK scheme as defined in Eq. (9a), with the state space identity matrix, and where $\boldsymbol{\kappa}$ is the vector of size $N_{\mathrm{RK}}N_x$ defined after Eq. (11). Looking first at the gradient with respect to the state variable, and using Eq. (12), we have

$$\nabla_{\mathbf{x}}\mathbf{f}_{\mathbf{A}} = \nabla_{\mathbf{x}}\mathbf{x} + \mathbf{b}\nabla_{\mathbf{x}}\boldsymbol{\kappa} = \mathbf{I}_x + \mathbf{b}\mathbf{G}^{-1}\nabla_{\mathbf{x}}\boldsymbol{\varphi}, \tag{22}$$

which yields the adjoint operator

$$\left(\nabla_{\mathbf{x}}\mathbf{f}_{\mathbf{A}}\right)^{\mathrm{T}} = \mathbf{I}_x + \left(\nabla_{\mathbf{x}}\boldsymbol{\varphi}\right)^{\mathrm{T}}\mathbf{G}^{-\mathrm{T}}\mathbf{b}^{\mathrm{T}}. \tag{23}$$

Let us consider an arbitrary vector $\mathbf{d} \in \mathbb{R}^{N_x}$; we have

$$\left(\nabla_{\mathbf{x}}\mathbf{f}_{\mathbf{A}}\right)^{\mathrm{T}}\mathbf{d} = \mathbf{d} + \left(\nabla_{\mathbf{x}}\boldsymbol{\varphi}\right)^{\mathrm{T}}\mathbf{G}^{-\mathrm{T}}\mathbf{b}^{\mathrm{T}}\mathbf{d}. \tag{24}$$





To avoid computing $\mathbf{G}^{-\mathrm{T}}$ explicitly, let us define the vector $\mathbf{z} \in \mathbb{R}^{N_x N_{\mathrm{RK}}}$ such that

$$\mathbf{G}^{\mathrm{T}}\mathbf{z} = \mathbf{b}^{\mathrm{T}}\mathbf{d}. \tag{25}$$

Because $\mathbf{G}^{\mathrm{T}}$ is upper triangular with diagonal entries of value 1, $\mathbf{z}$ is the solution of a linear system easily solvable iteratively, which stands as an adjoint/dual to the RK iterative construction. Hence, we finally obtain a formula and algorithm to evaluate

$$(\nabla_{\mathbf{x}}\mathbf{f_A})^{\mathrm{T}}\mathbf{d} = \mathbf{d} + (\nabla_{\mathbf{x}}\varphi)^{\mathrm{T}}\mathbf{z}, \tag{26}$$

which is key to computing Eq. (19). Indeed Eq. (19) now reads

$$\nabla_{\mathbf{x}_0}\mathcal{J} = -(\nabla_{\mathbf{x}_0}\mathbf{H}_0)^{\mathrm{T}}\mathbf{R}_0^{-1}\boldsymbol{\delta}_0 - \mathbf{Q}_1^{-1}\boldsymbol{\Delta}_1 - (\nabla_{\mathbf{x}_0}\varphi)^{\mathrm{T}}\mathbf{z}_0$$
$$\qquad - \nabla_{\mathbf{x}_0}\ln p(\mathbf{x}_0|\mathbf{A}), \tag{27a}$$
$$\nabla_{\mathbf{x}_k}\mathcal{J} = -(\nabla_{\mathbf{x}_k}\mathbf{H}_k)^{\mathrm{T}}\mathbf{R}_k^{-1}\boldsymbol{\delta}_k - \mathbf{Q}_{k+1}^{-1}\boldsymbol{\Delta}_{k+1} - (\nabla_{\mathbf{x}_k}\varphi)^{\mathrm{T}}\mathbf{z}_k$$

$$\qquad + \mathbf{Q}_k^{-1}\boldsymbol{\Delta}_k, \quad \text{for} \quad 1 \le k \le K-1, \tag{27b}$$
$$\nabla_{\mathbf{x}_K}\mathcal{J} = -(\nabla_{\mathbf{x}_K}\mathbf{H}_K)^{\mathrm{T}}\mathbf{R}_K^{-1}\boldsymbol{\delta}_K + \mathbf{Q}_K^{-1}\boldsymbol{\Delta}_K, \tag{27c}$$

where $\mathbf{z}_k$ is the iterative solution of the system $\mathbf{G}_k^{\mathrm{T}}\mathbf{z}_k = \mathbf{b}^{\mathrm{T}}\mathbf{Q}_{k+1}^{-1}\boldsymbol{\Delta}_{k+1}$ for $0 \le k \le K-1$.

Second, let us look at the gradient of $\mathcal{J}(\mathbf{A}, \mathbf{x}_{0:K})$ with respect to $\mathbf{A}$. From Eq. (21) and Eq. (12), and now considering variations with respect to $\mathbf{A}$, we obtain:

$$\nabla_{\mathbf{A}}\mathbf{f_A} = \nabla_{\mathbf{A}}\mathbf{x} + \mathbf{b}\nabla_{\mathbf{A}}\boldsymbol{\kappa} = \mathbf{b}\mathbf{G}^{-1}\nabla_{\mathbf{A}}\varphi, \tag{28}$$

which yields, using $\mathbf{z}$ as defined by Eq. (25),

$$\mathbf{d}^{\mathrm{T}}(\nabla_{\mathbf{A}}\mathbf{f_A}) = \mathbf{d}^{\mathrm{T}}\mathbf{b}\mathbf{G}^{-1}(\nabla_{\mathbf{A}}\varphi) = \mathbf{z}^{\mathrm{T}}(\nabla_{\mathbf{A}}\varphi). \tag{29}$$

For each $0 \le i < N_{\mathrm{RK}}$, let us introduce $\mathbf{r}_i = \mathbf{r}(\mathbf{x_i}) \in \mathbb{R}^{N_p}$, and let us denote $\mathbf{z}_i \in \mathbb{R}^{N_x}$ the subvector of $\mathbf{z}$ for the $i$-th block of the Runge-Kutta scheme. Then, we have for $0 \le n < N_x$ and $0 \le p < N_p$:

$$\left[\mathbf{z}^{\mathrm{T}}(\nabla_{\mathbf{A}}\varphi)\right]_{n,p} = \sum_{m=0}^{N_x-1}\sum_{i=0}^{N_{\mathrm{RK}}-1}[\mathbf{z}_i]_m \frac{\partial}{\partial A_{n,p}}\sum_{q=0}^{N_p-1}A_{m,q}[\mathbf{r}_i]_q$$
$$= \sum_{m=0}^{N_x-1}\sum_{i=0}^{N_{\mathrm{RK}}-1}\sum_{q=0}^{N_p-1}[\mathbf{z}_i]_m \delta_{m,n}\delta_{p,q}[\mathbf{r}_i]_q$$
$$= \sum_{i=0}^{N_{\mathrm{RK}}-1}[\mathbf{z}_i]_n[\mathbf{r}_i]_p = \sum_{i=0}^{N_{\mathrm{RK}}-1}\mathbf{z}_i\mathbf{r}_i^{\mathrm{T}}. \tag{30}$$

This is key to efficiently computing Eq. (20), which now reads

$$\nabla_{\mathbf{A}}\mathcal{J} = -\sum_{k=1}^{K}\sum_{i=0}^{N_{\mathrm{RK}}-1}\mathbf{z}_{k,i}\mathbf{r}_{k,i}^{\mathrm{T}} - \nabla_{\mathbf{A}}\ln p(\mathbf{A}), \tag{31}$$

where $\mathbf{z}_k$ is the solution of $\mathbf{G}_k^{\mathrm{T}}\mathbf{z}_k = \mathbf{b}^{\mathrm{T}}\mathbf{Q}_k^{-1}\boldsymbol{\delta}_k$. The index $k$ of $\mathbf{G}_k$ indicates that the operators defined in the entries of $\mathbf{G}$ are evaluated at $\mathbf{x}_k$.





### 2.5.2  Composition of integration steps

We now consider a resolvent which is the composition of $N_c^k \geq 2$ integration steps over $[t_k, t_{k+1}]$: $\mathbf{x}' = \mathbf{f}_{\mathbf{A}}^{N_c^k}(\mathbf{x})$ where $\mathbf{x}$ is an alias to $\mathbf{x}_k$. Let us first look at the gradient with respect to the state variable. Within the scope of this section, we define $\mathbf{x}_0 \equiv \mathbf{x}$ and for $1 \leq l \leq N_c^k$: $\mathbf{x}_l \equiv \mathbf{f}_{\mathbf{A}}(\mathbf{x}_{l-1})$. Hence, $\mathbf{x}' = \mathbf{x}_{N_c}$. We also define $\nabla_{\mathbf{x}_l}\mathbf{f}_{\mathbf{A}}$ to be the tangent linear of $\mathbf{f}_{\mathbf{A}}$ at $\mathbf{x}_l$. By Leibniz rule, we obtain

$$\left(\nabla_{\mathbf{x}}\mathbf{F}_{\mathbf{A}}^k\right)^{\mathrm{T}} = (\nabla_{\mathbf{x}_0}\mathbf{f}_{\mathbf{A}})^{\mathrm{T}}(\nabla_{\mathbf{x}_1}\mathbf{f}_{\mathbf{A}})^{\mathrm{T}}\cdots(\nabla_{\mathbf{x}_{N_c-1}}\mathbf{f}_{\mathbf{A}})^{\mathrm{T}}. \tag{32}$$

We can now apply Eq. (26) to each individual integration step and obtain for any $\mathbf{d} \in \mathbb{R}^{N_x}$

$$(\nabla_{\mathbf{x}_l}\mathbf{f}_{\mathbf{A}})^{\mathrm{T}}\mathbf{d} = \mathbf{d} + (\nabla_{\mathbf{x}_l}\boldsymbol{\varphi})^{\mathrm{T}}\mathbf{z}_l, \tag{33}$$

where $\mathbf{z}_l$ is the solution of

$$\mathbf{G}_l^{\mathrm{T}}\mathbf{z}_l = \mathbf{b}^{\mathrm{T}}\mathbf{d}. \tag{34}$$

Hence, to compute $\left(\nabla_{\mathbf{x}}\mathbf{F}_{\mathbf{A}}^k\right)^{\mathrm{T}}\cdot\mathbf{d}$, we define $\tilde{\mathbf{x}}_{N_c} = \mathbf{d}$ and for $N_c - 1 \geq l \geq 0$: $\tilde{\mathbf{x}}_l = (\nabla_{\mathbf{x}_l}\mathbf{f}_{\mathbf{A}})^{\mathrm{T}}\tilde{\mathbf{x}}_{l+1}$. This finally reads:

$$\tilde{\mathbf{x}}_l = \tilde{\mathbf{x}}_{l+1} + (\nabla_{\mathbf{x}_l}\boldsymbol{\varphi}_l)^{\mathrm{T}}\mathbf{z}_l, \tag{35}$$

for $N_c - 1 \geq l \geq 0$, where $\mathbf{z}_l$ is the solution of

$$\mathbf{G}_l^{\mathrm{T}}\mathbf{z}_l = \mathbf{b}^{\mathrm{T}}\tilde{\mathbf{x}}_{l+1}. \tag{36}$$

To compute the key terms in the gradients Eq. (19), $\mathbf{d}$ must be chosen to be $\mathbf{Q}_{k+1}^{-1}\boldsymbol{\Delta}_{k+1}$ where $0 \leq k \leq K - 1$ and

$$\left(\nabla_{\mathbf{x}}\mathbf{F}_{\mathbf{A}}^k\right)^{\mathrm{T}}\mathbf{Q}_{k+1}^{-1}\boldsymbol{\Delta}_{k+1} = \tilde{\mathbf{x}}_0. \tag{37}$$

Second, we look at the gradient with respect to $\mathbf{A}$. In this case, the application of the Leibniz rule yields

$$\nabla_{\mathbf{A}}\mathbf{F}_{\mathbf{A}}^k = \sum_{l=0}^{N_c-1}(\nabla_{\mathbf{x}_{N_c-1}}\mathbf{f}_{\mathbf{A}})\cdots(\nabla_{\mathbf{x}_{l+1}}\mathbf{f}_{\mathbf{A}})\nabla_{\mathbf{A}}\mathbf{f}_{\mathbf{A}}(\mathbf{x}_l)$$
$$= \sum_{l=0}^{N_c-1}\left(\nabla_{\mathbf{x}_{l+1}}\mathbf{f}_{\mathbf{A}}^{N_c-l-1}\right)\nabla_{\mathbf{A}}\mathbf{f}_{\mathbf{A}}(\mathbf{x}_l), \tag{38}$$

where $\nabla_{\mathbf{x}_{l+1}}\mathbf{f}_{\mathbf{A}}^{N_c-l-1} = (\nabla_{\mathbf{x}_{N_c-1}}\mathbf{f}_{\mathbf{A}})\cdots(\nabla_{\mathbf{x}_{l+1}}\mathbf{f}_{\mathbf{A}})$. But $\nabla_{\mathbf{A}}\mathbf{f}_{\mathbf{A}}(\mathbf{x}_l)$, which focuses on a single integration step, is given by Eq. (28):

$$\nabla_{\mathbf{A}}\mathbf{f}_{\mathbf{A}}(\mathbf{x}_l) = \mathbf{b}\mathbf{G}_l^{-1}\nabla_{\mathbf{A}}\boldsymbol{\varphi}_l \tag{39}$$

and from Eq. (29):

$$\mathbf{d}^{\mathrm{T}}\nabla_{\mathbf{A}}\mathbf{f}_{\mathbf{A}}(\mathbf{x}_l) = \mathbf{z}_l^{\mathrm{T}}\left(\nabla_{\mathbf{A}}\boldsymbol{\varphi}_l\right). \tag{40}$$

As a result, we obtain:

$$\nabla_{\mathbf{A}}\mathcal{J} = -\sum_{l=0}^{N_c-1}\sum_{k=1}^{K}\sum_{i=0}^{N_{\mathrm{RK}}-1}\mathbf{z}_{k,l,i}\mathbf{r}_{k,l,i}^{\mathrm{T}} - \nabla_{\mathbf{A}}\ln p(\mathbf{A}), \tag{41}$$

where $\mathbf{z}_{k,l}$ is the solution of

$$\mathbf{G}_{k,l}^{\mathrm{T}}\mathbf{z}_{k,l} = \mathbf{b}^{\mathrm{T}}\left(\nabla_{\mathbf{x}_{l+1}}\mathbf{f}_{\mathbf{A}}^{N_c-l-1}\right)^{\mathrm{T}}\mathbf{Q}_k^{-1}\boldsymbol{\delta}_k. \tag{42}$$

All of these results, Eqs. (27,31,37,41), allow to efficiently compute the gradients of the cost function $\mathcal{J}(\mathbf{A},\mathbf{x}_{0:K})$ with respect to both $\mathbf{A}$ and $\mathbf{x}_{0:K}$. Note, however, that they have been derived under the simplifying assumption that $\phi_{\mathbf{A}}$ is given by Eq. (2) with a traditional matrix multiplication between $\mathbf{A}$ and $\mathbf{r}(\mathbf{x})$, but not by the compact Eq. (7). When relying on homogeneity and/or locality, the calculation of the gradient with respect to $\mathbf{A}$ follows the principle described above but requires further adaptations, which can be derived using Eq. (A2), with the asset of strongly reducing the computational burden.

## 3 Discussion of the theoretical points

In this section, we discuss about the optimisation of the cost function $\mathcal{J}(\mathbf{A},\mathbf{x}_{0:K})$. The goal is either to reconstruct an ODE for the reference model, characterised by the coefficients $\mathbf{A}$ through $\phi_{\mathbf{A}}$, or to build a surrogate model of it. The estimation of $\mathbf{x}_{0:K}$ is then accessory, even though critical to the estimation of $\mathbf{A}$. The alternative would have been to consider the estimation of $\mathbf{x}_{0:K}$ as the primary problem, under model error of a prescribed ODE form, the estimation of $\mathbf{A}$ being accessory. In this latter case, one may benefit from an informative prior pdf $p(\mathbf{A})$. The prior pdf $p(\mathbf{A})$ can also be used to encode a partially specified model or any prior knowledge on the reference model.

### 3.1 Numerical optimisation: issues and solutions

The success of the optimisation of $\mathcal{J}(\mathbf{A},\mathbf{x}_{0:K})$ depends above all on the ability to robustly evaluate it. In particular, it depends on the stability of the numerical integration scheme $\mathbf{x}' = \mathbf{f}_{\mathbf{A}}(\mathbf{x})$. In this paper, we chose to rely on one-step explicit schemes which are much simpler to describe and efficient to integrate (a family to which the RK schemes of any $N_{\mathrm{RK}}$ belong). These schemes are 0-stable, which means that the finite time error growth goes to zero as the integration step goes to zero. But, as a major drawback, they have a limited absolute (or A–)stability domain (see e.g., chapters 5 and 6 of Gautschi, 2012). For a given state trajectory, there exists a stability domain $\mathcal{D}_{\mathrm{s}} \in \mathbb{R}^{N_x \times N_p}$ out of which the evaluation of $\mathcal{J}(\mathbf{A},\mathbf{x}_{0:K})$ is hazardous. A failure to evaluate the cost function depends on the number of integration steps. Indeed, instabilities are to increase exponentially with $N_{\mathrm{c}}^k$.

This tells that, in the absence of a strong prior $p(\mathbf{A})$, it is safer to start with $\mathbf{A} = \mathbf{0}$ likely to lie close to $\mathcal{D}_{\mathrm{s}}$. Alternatively, if stability constraints are known about $\mathbf{A}$, they could be encoded in $p(\mathbf{A})$. It also tells that we should strike an empirical compromise between the composition numbers $N_{\mathrm{c}}^k$ and the easiness to evaluate $\mathcal{J}(\mathbf{A},\mathbf{x}_{0:K})$. In the one hand, the larger $N_{\mathrm{c}}^k$ the more iterates of $\mathbf{A}$ in the optimisation must be kept confined in $\mathcal{D}_{\mathrm{s}}$. On the other hand, the longer $N_{\mathrm{c}}^k$ the broader the class





of achievable resolvents, and hence the ability to build a good surrogate. Moreover, the higher the stability of the integration scheme, the larger $\mathcal{D}_\mathrm{s}$, and the easier the optimisation in spite of an increase in its numerical cost.

The longer $K$ is, the more observations are available to constraint the problem. However, the longer $K$ the higher the chances of having a significant instability: a successful integration typically decreases exponentially with the length $K$.

This stability issue can be somehow alleviated by normalising the observations $\mathbf{y}_k$ by their mean and variance in order to avoid excessively large value ranges of the regressors. This will not change the fundamental stability of the schemes, yet may delay the occurrences of instabilities due to the nonlinear terms.

Moreover, instabilities can significantly be mitigated by replacing the monomials with smoothed or truncated ones:

$$\mathbf{r}(\mathbf{x}) = \left[ \left\{ \zeta(\tilde{x}_n)\zeta(\tilde{x}_m) \right\}_{(n,m)\in\widetilde{\mathcal{P}}} \right]. \tag{43}$$

One can for instance choose $\zeta(x) = \lambda\tanh(x/\lambda)$, in order to cut-off too large values of $|x|$ and hence delay the growth of instabilities. The parameter $\lambda > 0$ is roughly chosen as the typical maximum amplitude of $|x|$. If $\tanh$ is deemed to be numerically too costly, one can choose instead $\zeta(x) = -\lambda 1_{]-\infty,-\lambda]} + x1_{[-\lambda,\lambda]} + \lambda 1_{[\lambda,+\infty[}$. This latter change of variables is the one implemented in Sect. 4, together with the normalisation. These tricks often turn critical in the first iterates of the optimisation as the estimate of $\mathbf{A}$ progressively migrates to the A-stability domain.

## 15   3.2   Connection and analogies with deep learning architectures

It has recently been advocated that residual deep learning architectures of neural networks can roughly be interpreted as dynamical systems (e.g., E, 2017; Chang et al., 2018). Each layer of the network contributes marginally to the output, so that there exists an asymptotic continuum limit representation of the neural network. Furthermore, as mentioned in the introduction, Wang and Lin (1998); Fablet et al. (2018) have shown that the architecture of the network can follow that of an integration

scheme.

By contrast, we have started here from a pure dynamical system standpoint, and proposed to use data assimilation techniques. In order to explore complex model resolvents, applied to each interval $[t_k, t_{k+1}]$ between observations, we need the composition of $N_\mathrm{c}$ integration steps. In particular, this allows for the resolvent to exhibit more realistic long-range correlations. Even when using a reasonably small stencil, long-correlations will arise as a result of the integration steps. Nonetheless the stencil might not

be too small so as to model discretised higher-order differential operators. As noted by Abarbanel et al. (2018), each application of $\mathbf{f_A}$ could be seen as a layer of the neural network. Moreover, within such layer, there are sublayers corresponding to the steps of the integration scheme. The larger $N_\mathrm{c}$ is the deeper this network is and the richer the class of resolvents to optimise on.

Following this analogy, the analysis step where $\mathcal{J}(\mathbf{A}, \mathbf{x}_{0:K})$ is optimised can be called the training phase. Backpropagation in the network, as coined in machine learning (Goodfellow et al., 2016), corresponds to the computation of the gradient of the

network with respect to $\mathbf{A}$ and of the model adjoint derived in Sect. 2. This a shortcut for the use of machine learning software libraries such as TensorFlow or PyTorch.





Because of our complete control of the backpropagation, we hope for a gain in efficiency. However, our method does not have the flexibility of the use of deep learning through established tools. For instance, adding extra parameters, adaptive batch normalisation, dropouts are not granted in our approach.

Convolutional layers play the role of localisation in neural architecture. In our approach this role is played by the locality assumption and its stencil prescription. Recall that a tight stencil does not prevent longer-range correlation that are built up through the integration scheme and their composition. This is similar to stacking several convolutional layers to account for multiple scales from the reference model which the neural network is meant to learn from.

Finally, we note that, as opposed to deep learning strategies with a huge amount of weights to estimate, we have reduced the number of control variables (i.e. $\mathbf{A}$) as much as possible.

## 4  Numerical illustrations

### 4.1  Models setup and forecast skill

In this section, we shall consider four low-order chaotic models that will serve as reference model:

1. the L63 model as defined by the ODEs:

$$\frac{\mathrm{d}x_0}{\mathrm{d}t} = \sigma(x_1 - x_0), \tag{44a}$$

$$\frac{\mathrm{d}x_1}{\mathrm{d}t} = \rho x_0 - x_1 - x_0 x_2, \tag{44b}$$

$$\frac{\mathrm{d}x_2}{\mathrm{d}t} = \rho x_0 x_1 - \beta x_2, \tag{44c}$$

with the canonical values $(\sigma, \rho, \beta) = (10, 28, 8/3)$. Its Lyapunov time[1] is about $1.10$. This model is introduced for benchmarking with e.g., Fablet et al. (2018). It is integrated relying on an RK4 scheme with $\delta t_{\mathrm{r}} = 0.01$ as the integration time-step.

2. the L96 model as defined by ODEs defined over a periodic domain of variables indexed by $n = 0, \ldots, N_x - 1$ where $N_x = 40$:

$$\frac{\mathrm{d}x_n}{\mathrm{d}t} = (x_{n+1} - x_{n-2})x_{n-1} - x_n + F, \tag{45}$$

where $x_{N_x} = x_0, x_{-1} = x_{N_x-1}, x_{-2} = x_{N_x-2}$, and $F = 8$. This model is an idealised representation of a one-dimensional latitude band of the Earth atmosphere for which localisation approaches can be tested. Its Lyapunov time is $0.60$. It is integrated with an RK4 scheme and $\delta t_{\mathrm{r}} = 0.05$.

3. the KS model, as defined by the partial differential equations:

$$\frac{\partial x}{\partial t} = -x \frac{\partial x}{\partial \alpha} - \frac{\partial^2 x}{\partial \alpha^2} - \frac{\partial^4 x}{\partial \alpha^4}, \tag{46}$$

---

[1]The Lyapunov time is defined as the inverse of the first Lyapunov exponent, which corresponds to a growth of the error by $e$.





over the periodic domain $\alpha \in [0, 32\pi]$ on which we apply a spectral decomposition with $N_x = 128$ modes. The Lyapunov time of our KS model is 10.2 time units. This model is of interest because, even though it has dynamical properties comparable to that of L96, it is much steeper so that much more stringent numerical integration schemes are required to efficiently integrate it. It is integrated using the EDTRK4 scheme (Kassam and Trefethen, 2005) and $\delta t_r = 0.05$.

4. the two-scale Lorenz model (L05III, Lorenz, 2005) is given by the two-scale ODEs:

$$\frac{\mathrm{d}x_n}{\mathrm{d}t} = \psi_n^+(\mathbf{x}) + F - h\frac{c}{b}\sum_{m=0}^{9} u_{m+10n}, \tag{47a}$$

$$\frac{\mathrm{d}u_m}{\mathrm{d}t} = \frac{c}{b}\psi_m^-(b\mathbf{u}) + h\frac{c}{b}x_{m/10}, \tag{47b}$$

$$\psi_n^\pm(\mathbf{x}) = x_{n\mp1}(x_{\pm1} - x_{n\mp2}), \tag{47c}$$

for $n = 0, \ldots, N_x - 1$ with $N_x = 36$ and $m = 0, \ldots, N_u - 1$ with $N_u = 360$. The indices apply periodically over their domain; $m/10$ is the integer division of $m$ par 10. We use the original values for the parameters: $c = 10$ for the timescale ratio, $b = 10$ for the space-scale ratio, $h = 1$ for the coupling, and $F = 15$ for the forcing. The Lyapunov time of the model is 0.72.

This model is of interest because the variable $\mathbf{u}$ is meant to represent unresolved scales and hence model error when only considering the large scale variable $\mathbf{x}$. For this reason, it has been used in data assimilation papers focusing on estimating model error (e.g., Mitchell and Carrassi, 2015; Pulido et al., 2018). It is integrated with an RK4 scheme and $\delta t_r = 0.005$ since it is steeper than the L96 model.

The numerical experiments consist of three main steps. First, the truth is generated; i.e. a trajectory of the reference model is computed. The reference model equations are supposed to be unknown, but the trajectory is observed through Eq. (13) to generate the observation vector sequence $\mathbf{y}_{0:K}$.

Next, estimators of the ODEs model and state trajectory $\mathbf{x}_{0:K}$ are learned by minimising the cost function $\mathcal{J}(\mathbf{A}, \mathbf{x}_{0:K})$. We choose to minimise it using the quasi-Newton BFGS algorithm, which critically relies on the gradients obtained in Sect. 2. The default choices for the initial ODEs model are $\mathbf{A} = 0$ and $\mathbf{x}_{0:K}$ defined as the space-wise linear interpolation of $\mathbf{y}_{0:K}$. Note that the minimisation could converge to a local minimum, which may or may not yield satisfactory estimators.

Finally, we can make forecasts using the tentative optimal ODEs model $\mathbf{A}^\star$ obtained from the minimisation. With a view

to compare it to the reference model used to generate the data, we will consider a set of forecasts with (approximately) independent initial conditions. Both the reference model and the surrogate one will be forecasted from these initial conditions. The departure from their trajectories, as measured by a root mean square error (RMSE) over the observed variables, will be computed for several forecast lead times. The RMSE is then averaged over all the initial conditions. We will also display the state trajectories of the reference and surrogate models starting from one of the initial conditions.

The integration time step of the truth (reference model) is $\delta t_r$ over the time window $[t_0, t_K]$. This parameter only matters for the reference model integration since only the training time steps $t_{k+1} - t_k$ and the output of the model $\mathbf{y}_{0:K}$ (which may include knowledge of the observation operator) are known to the observer.



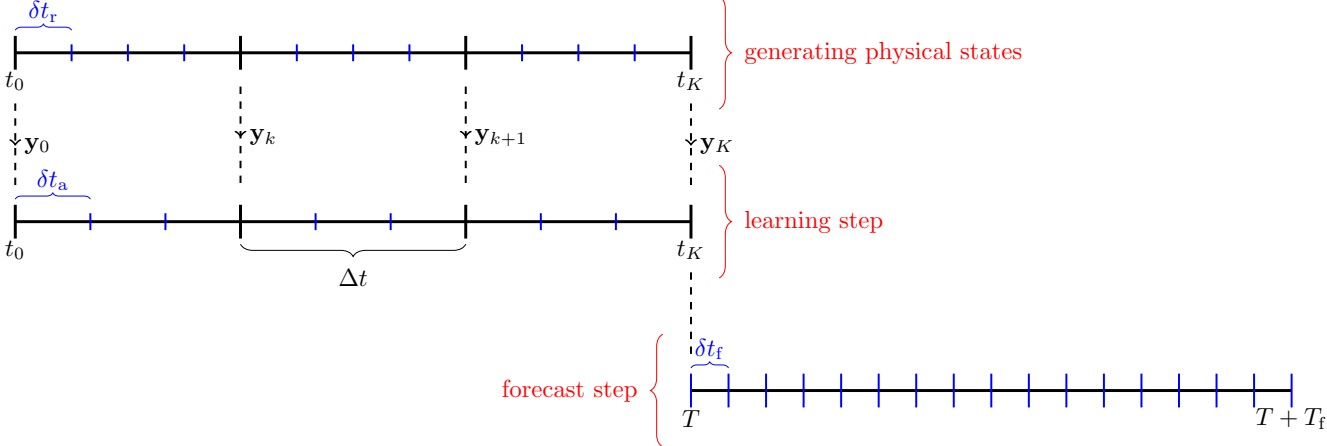

**Figure 2.** Schematic of the three steps of the experiments, with the associated time steps (see main text). The beginning of the forecast window may or may not coincide with the end of the learning window. The lengths of the segments $\delta t_\mathrm{r}$, $\delta t_\mathrm{a}$, and $\delta t_\mathrm{f}$ are arbitrary in this schematic.

The integration time step of the surrogate model within this learning time window $[t_0, t_K]$ is $\delta t_\mathrm{a}$. It is assumed to be an integer divisor of the training time step $\Delta t = t_{k+1} - t_k$ supposed to be constant, i.e. $\Delta t / \delta t_\mathrm{a}$ is a constant integer, the number of compositions $N_\mathrm{c}$, and that is why the index $k$ on $N_\mathrm{c}^k$ has been dropped. The integration time step of the surrogate model within the forecast time window $[T, T + T_\mathrm{f}]$ is denoted $\delta t_\mathrm{f}$. Note that $\delta t_\mathrm{a}$ and $\delta t_\mathrm{f}$ can be distinct. They are critical to the stability of

the learning and the forecast step, respectively.

The three steps of the numerical experiments are depicted in Fig. 2. Except when explicitly mentioned, the prior $p(\mathbf{A})$ is disregarded, which means that no regularisation on $\mathbf{A}$ is introduced.

### 4.2 Inferring the dynamics from dense and noiseless observations of perfectly identifiable models

In the first couple of experiments, we consider a densely observed[2] reference model with noiseless observations. In this case,

$\mathbf{R}_k \equiv \mathbf{0}$ and a uniform rescaling of the $\mathbf{Q}_k$, all chosen to be $\mathbf{I}_x$, is irrelevant, assuming the prior $p(\mathbf{y}_0, \mathbf{A})$ can be neglected which is assumed here and is generally true for large $K$. Moreover, we use the same numerical scheme and the same integration time step to generate the reference model trajectory as the one used by the surrogate model. In principle, we should be able to retrieve the reference model, since the reference is identifiable among all the possible surrogate models.

Let us first experiment with the L63 model, using an RK4 integration scheme, with $\Delta t = 0.01$ and $K = 10^4$. We have

$N_x = 3$ and $N_p = 10$. We choose $\delta t_\mathrm{a} = \delta t_\mathrm{f} = 0.01$. A convergence to the highest possible precision is achieved after about 120 iterations. The cost function value reaches 0 to machine precision at $\mathbf{A}^\star$. The estimated $\mathbf{A}$ is given by $\mathbf{A}_\mathrm{a} = \mathbf{A}^\star / \delta t_\mathrm{a}$, because,

---

[2]We choose the qualifier *densely observed* instead of *fully observed* because there is no way to tell from the observations alone if the reference model is fully observed.



as mentioned above, the optimised $\mathbf{A}$ matrix absorbs the time step. The accuracy of $\mathbf{A}_\mathrm{a}$ is measured by the uniform norm $\|\mathbf{A}_\mathrm{a} - \mathbf{A}_\mathrm{r}\|_\infty$, i.e. the absolute values of the entries of the difference $\mathbf{A}_\mathrm{a} - \mathbf{A}_\mathrm{r}$, where $\mathbf{A}_\mathrm{r}$ is the matrix of the flow rate of L63 (including the zero coefficients). We obtain $\|\mathbf{A}_\mathrm{a} - \mathbf{A}_\mathrm{r}\|_\infty = 2.00 \times 10^{-10}$. To compute the RMSE as a function of the forecast lead time, we average over $N_\mathrm{e} = 10^3$ runs (each one starting from a difference initial condition). The RMSE (not shown) starts significantly diverging from 0 after 16 Lyapunov time units and reaches a saturation for a lead time of 23.

A similar experiment is carried out with the L96 model, using an RK4 integration scheme, with $\Delta t = 0.05$ and $K = 50$. We choose here to implement the locality and homogeneity assumptions. The stencil has a width of 5 (i.e. the local grid points with 2 points on its left and 2 points on its right). We have $N_x = 40$, $N_p = 161$ and $N_\mathrm{a} = 18$. We choose $\delta t_\mathrm{a} = \delta t_\mathrm{f} = 0.05$. The minimisation stops after about 30 iterations at the highest possible accuracy and the cost function value reaches 0 to machine precision. The main coefficients of the L96 model (forcing $F$, advections terms, dissipation) are retrieved with a precision of at least $3.61 \times 10^{-8}$.

To compute the RMSE as a function of the forecast lead time, we average over $N_\mathrm{e} = 10^3$ runs. The RMSE starts significantly diverging from 0 after 12 time units and reaches a saturation for a lead time of 25.

### 4.3 Inferring the dynamics from dense and noiseless observations of a non-identifiable model

In this second couple of experiments, we consider again a densely observed reference model with noiseless observations. The reference model trajectory is generated by the L96 model ($N_x = 40$) integrated with the RK4 scheme, with $\Delta t = 0.05$ and $K = 50$.

As opposed to the reference model, in these non-identifiable model experiments, the surrogate model is based on the RK2 scheme, with $N_\mathrm{c}$ compositions. We choose to implement the locality and homogeneity assumptions, with a stencil of width 5. We have $N_p = 161$ and $N_\mathrm{a} = 18$. We choose $\delta t_\mathrm{a} = \delta t_\mathrm{f} = \Delta t / N_\mathrm{c}$. In all cases, the convergence is reached within a few dozens of iterations. The error on the coefficients of $\mathbf{A}$ (i.e. $\|\mathbf{A}_\mathrm{a} - \mathbf{A}_\mathrm{r}\|_\infty$) is about $4 \times 10^{-2}$ but with the dominant contribution from $F$. The RMSE as a function of the forecast lead time is computed for $N_\mathrm{c} = 1, 2, 3, 4, 5$ and shown in Fig. 3. The error is reduced as $N_\mathrm{c}$ is increased. But the improvement saturates at about $N_\mathrm{c} = 5$.

Figure 4 shows the trajectories of the reference and surrogate models starting from the same initial condition, as well as their difference, as a function of the forecast lead time. Their divergence becomes significant after 4 Lyapunov times and saturates after 8 Lyapunov times.

Next, the reference model trajectory is generated by the KS model ($N_x = 128$) integrated with the ETDRK4 scheme, with $\Delta t = 0.05$ and $K = 50$. We choose to implement the locality and homogeneity assumptions, with a stencil of width 9. The surrogate model is based on the RK4 scheme, with $N_\mathrm{c} = 2$ compositions. Note that in this experiment, the reference and surrogate models and their integration schemes significantly differ. We have $N_p = 769$ and $N_\mathrm{a} = 45$. We choose $\delta t_\mathrm{a} = \Delta t / N_\mathrm{c}$ and $\delta t_\mathrm{f} = 10^{-3}$. The forecast time step $\delta t_\mathrm{f}$ is somehow smaller than $\delta t_\mathrm{a}$ because the KS equations are stiff and so will the surrogate model. This emphasises once again that we have learned about the intrinsic flow rate of the reference model, and not a resolvent thereof. Alternatively, we could use a more robust integration scheme than RK4 such as ETDRK4 for the forecast.





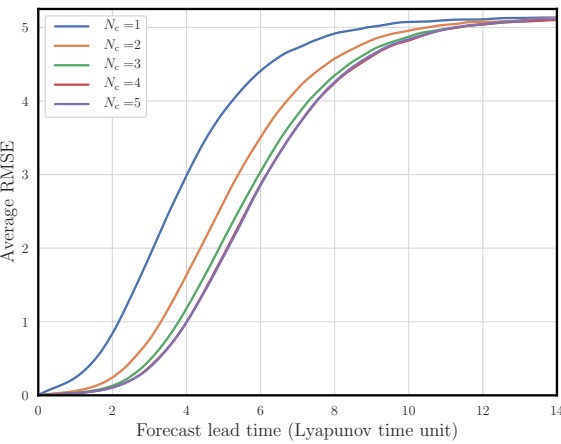

**Figure 3.** Average RMSE of the surrogate model (L96 with an RK2 structure) compared to the reference model (L96 with an RK4 integration scheme) as a function of the forecast lead time (in Lyapunov time unit) for an increasing number of compositions.

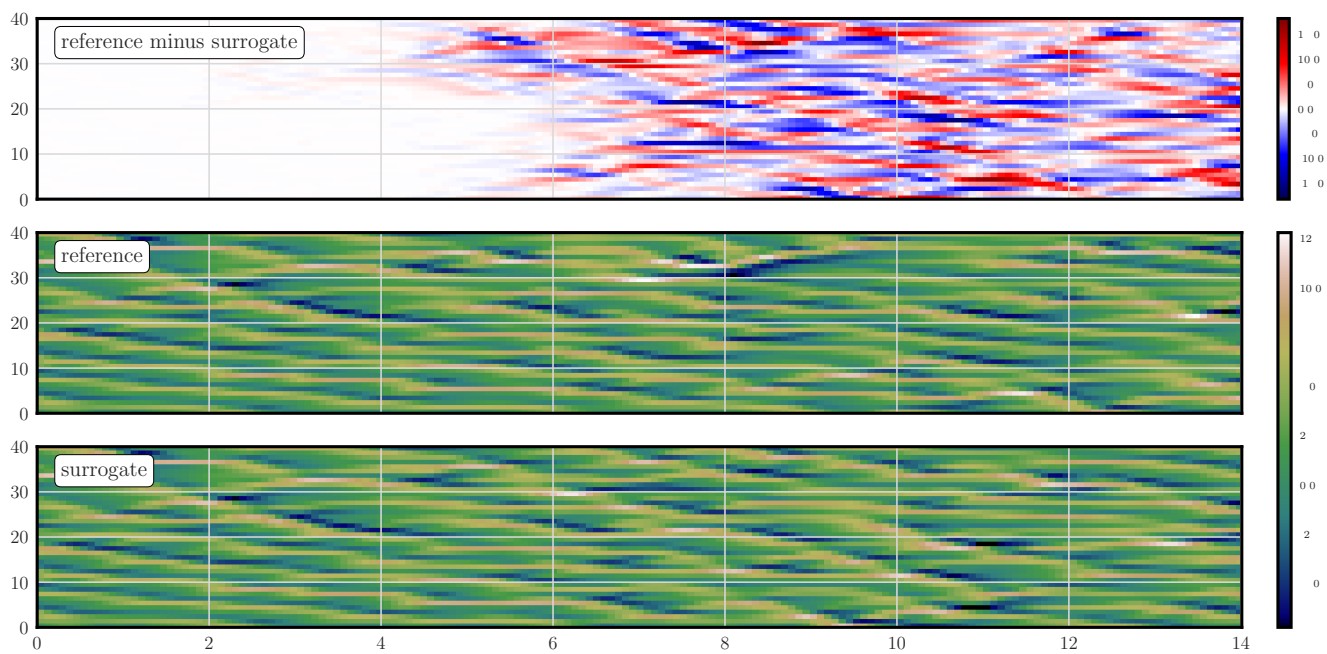

**Figure 4.** Density plot of the L96 reference and surrogate model trajectories, as well as their difference trajectory, as a function of the forecast lead time (in Lyapunov time unit). The observations are noiseless and dense; the model is not identifiable.



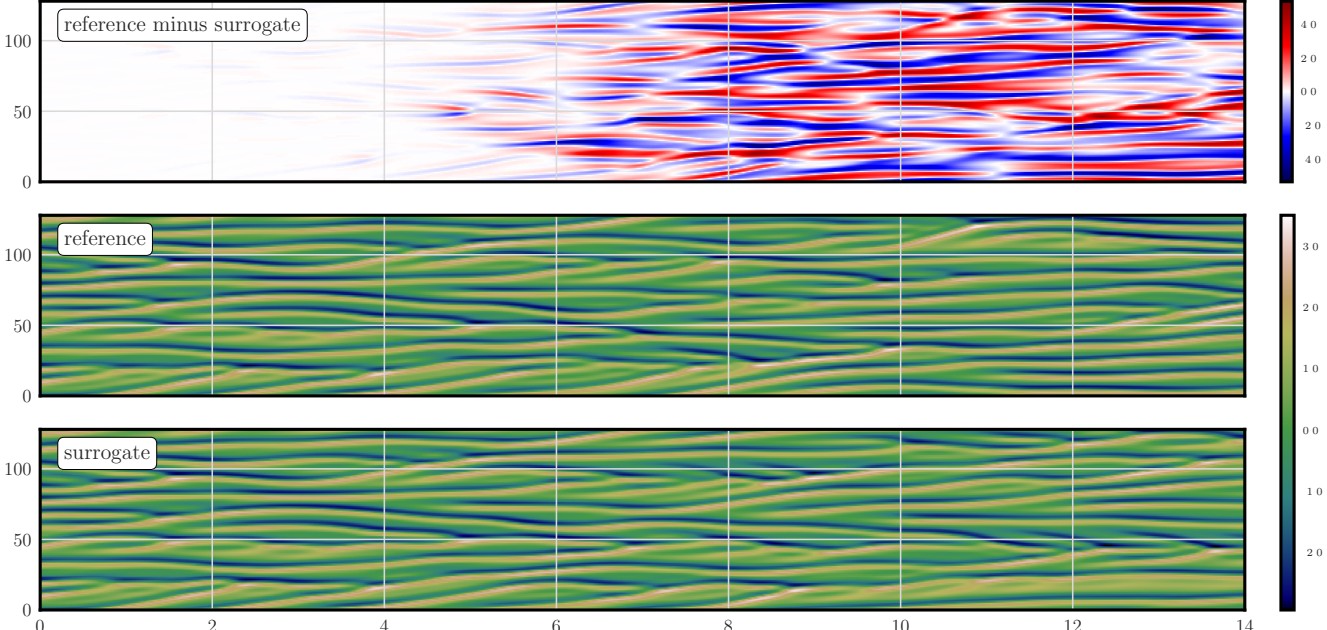

**Figure 5.** Density plot of the KS reference and surrogate model trajectories, as well as their difference trajectory, as a function of the forecast lead time (in Lyapunov time unit). The observations are noiseless and dense; the model is not identifiable.

Figure 5 shows the trajectories of the reference and surrogate models starting from the same initial condition, as well as their difference, as a function of the forecast lead time, for a stencil of $9$. Their divergence becomes significant after $4$ Lyapunov times and saturates after $8$ Lyapunov times.

To check whether the PDE of the KS model could be retrieved in spite of the differences in the method of integrations and representations, we have computed a Taylor expansion of all monomials in the surrogate ODEs flow rate up to order $4$ so as to obtain an approximately PDE equivalent. The coefficients of this PDE (up to order $4$ in the expansion) are displayed in Fig. 6 and compared to the coefficients of the reference model's PDE. The match is good and the terms $-x\partial_\alpha x$, $-\partial_\alpha^2 x$ and $-\partial_\alpha^4 x$ are correctly identified as the dominant ones. Nonetheless, there are three non-negligible coefficients for higher-order terms that may have been generated by the Taylor expansion, or may originate from a degeneracy among the higher-order operators, or may simply be identified with a shortcoming of our specific ODE representation.

## 4.4 Inferring the dynamics from partial and noisy observations

We come back to the L96 model which is densely observed but with noisy observations that are generated using an independently identically distributed normal noise. The surrogate model is based on an RK4 scheme, $N_{\mathrm{c}} = 1$ and a stencil of length $5$, which makes the reference model identifiable. In this case, the outcome theoretically depends on the choice for $\mathbf{R}_k$ and $\mathbf{Q}_k$, as



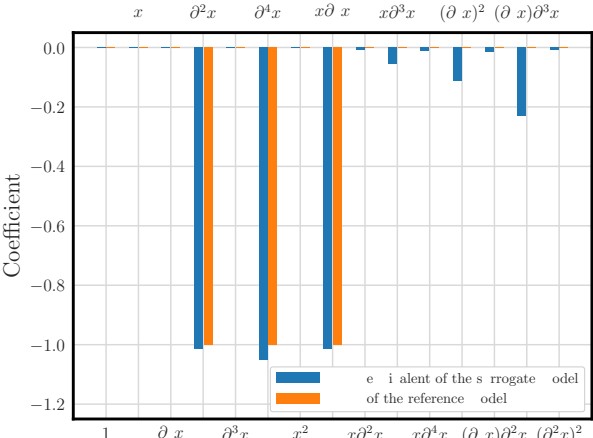

**Figure 6.** Coefficients of the surrogate PDE model (blue) resulting from the expansion of the surrogate ODEs, and compared to the reference PDE's coefficients (orange).

Eq. (17) is now used instead of Eq. (18). For the sake of simplicity, we have chosen them to be of the scalar form $\mathbf{R}_k \equiv \sigma_{\mathrm{y}}^2 \mathbf{I}_y$ and $\mathbf{Q}_k \equiv q\mathbf{I}_x$. In this synthetic experiments, $\sigma_{\mathrm{y}}$ is supposed to be known. However, $q$ is not. We only have a qualitative view on the potential mismatch between the reference and the surrogate model. A Gaussian additive noise might not even be the best statistical model for such error. The optimal value of $q$ could be determined using an empirical Bayes approach based on, for

instance, the expectation-maximisation technique in order to determine the maximum a posteriori of the conditional density of $q$ (see e.g., Dreano et al., 2017; Pulido et al., 2018). However, this would make us deviate too much from the objective of this work. In the following we have chosen values of $q$ that yield near to optimal skill scores (typically $q \approx 10^{-3}\sigma_y^2$).

Moreover, we have chosen the relatively small $K = 50$. It must be beneficial to increase $K$ but this would force us to address issues relative to weak constraint 4D-Var optimisation for long time windows, a topic which is also beyond the scope of this

paper. Preliminary results on this topic are nonetheless discussed later in this section.

Figure 7 shows the forecast skill of the surrogate model as a function of the forecast lead time, and for increasing noise in the observations. Even though, in this configuration, the model is identifiable, the reference value $\mathbf{A}_0$ for $\mathbf{A}$ may not correspond to a minimum of the cost function. The cost function might have several local minima. As a consequence, there is no guarantee, starting from a non-trivial initial value for $\mathbf{A}$, that the model will be identified. Indeed, as seen in Fig. 7, the forecast skill

degrades significantly as the observation error standard deviation is increased.

This is confirmed by Fig. 8 where the precision in identifying the model, measured by either the spectral norm $\|\mathbf{A}_0 - \mathbf{A}\|_2$ or the uniform norm $\|\mathbf{A}_0 - \mathbf{A}\|_\infty$ are plotted as functions of the observation error standard deviation.





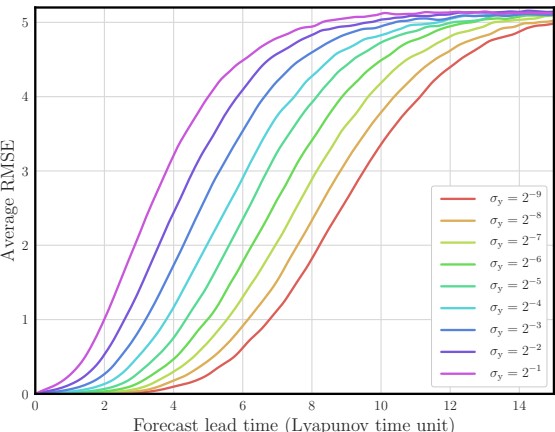

**Figure 7.** Average RMSE of the surrogate model (L96 with an RK4 structure) compared to the reference model (L96 with an RK4 integration scheme) as a function of the forecast lead time (in Lyapunov time unit) for a range of observation error standard deviation $\sigma_y$.

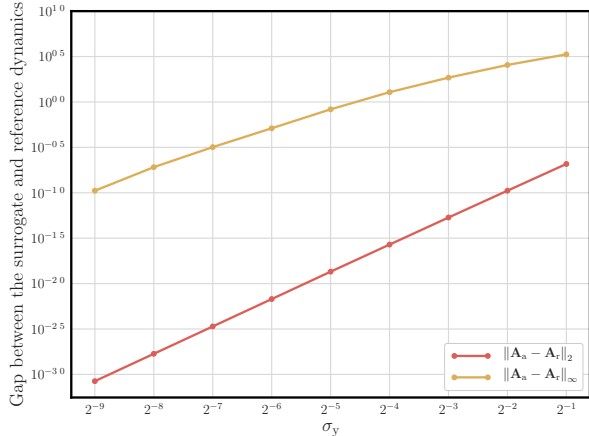

**Figure 8.** Gap between the surrogate (L96 with an RK4 structure) and the (identifiable) reference dynamics (L96 with an RK4 integration scheme) as a function of the observation error standard deviation $\sigma_y$. Note the use of logarithmic scales.





Using the same setup, we have also reduced the number of observations. The observations are regularly spaced and shifted by one grid cell at each observation time step. The initial $\mathbf{A}$ in the optimisation remains $\mathbf{0}$ while the the initial state $\mathbf{x}_{0:K}$ is taken as a cubic spline interpolation of the observations over the whole surrogate model grid.

If the observations are noiseless, the reference model is easily retrieved to a high precision down to a density of $1$ site over

$4$. If the observations are noisy, the performance slowly degrades down as the density is decreased down to about $1$ site over $4$, below which the minimisation, trapped in a deceiving local minimum, yields an improper surrogate model.

Finally, we would like to point out that in the case of noiseless observations, the performance little depends on the length of the learning window, beyond a relatively short length, typically $K = 50$. However, in presence of noisy observations, the overall performance improves with longer $K$, as expected since the information content of the observations linearly increases

with the length of the window.

## 4.5   Inferring reduced dynamics of a multiscale model

In this experiment, we consider the L05III model. With the locality and the homogeneity assumptions, the scalability is typically linear with the size of the system, and we actually consider the 10-fold model where $N_x = 360$ and $N_u = 3600$ to demonstrate that no issues was encountered when scaling up the method. The large scale variable $\mathbf{x}$ of the reference model is noiselessly and

fully observed over a short learning window ($K = 50$) whereas the short scale variable $\mathbf{u}$ is not observed. The surrogate model is based on the RK4 scheme and $N_c = 2$ compositions. We choose to implement the locality and homogeneity assumptions, with a stencil of width $5$. We have $N_p = 161$ and $N_a = 18$. We choose $\delta t_a = \delta t_f = \Delta t / N_c$.

Figure 9 shows the trajectories of the reference and surrogate models starting from the same initial condition, as well as their difference, as a function of time.

The emergence of error, i.e. the divergence from the reference, appears as long space-time stripes. We argue that they result from the emergence of subscale perturbations that are not properly represented by the surrogate model. Reciprocally there are long lasting stripes of low error not yet impacted by subscale perturbations. As expected and similarly to the L96 model, the perturbations are transported eastward as shown by the upward tilt of the stripes in Fig. 9. Clearly, in this case, a flow rate of the form Eq. (2) could be insufficient. Adding a stochastic parametrisation with parameters additionally inferred might offer

a solution, as in Pulido et al. (2018). Because of this mixed performance, the RMSE slowly degrades (compared to the other experiments reported so far) with the increase of the forecast lead time (not shown).

## 5   Conclusions

We have proposed to infer the dynamics of a reference model from its observation using Bayesian data assimilation, which is a new and original scope for data assimilation. Over a given training time window, the control variables are the state trajectory

and the coefficients of an ODE representation for the surrogate model. We have chosen the surrogate model to be the composition of an explicit integration scheme (Runge-Kutta typically) applied to this ODE representation. Time-invariance, space homogeneity and locality of the dynamics can be enforced making the method suitable for high-dimensional systems. The cost



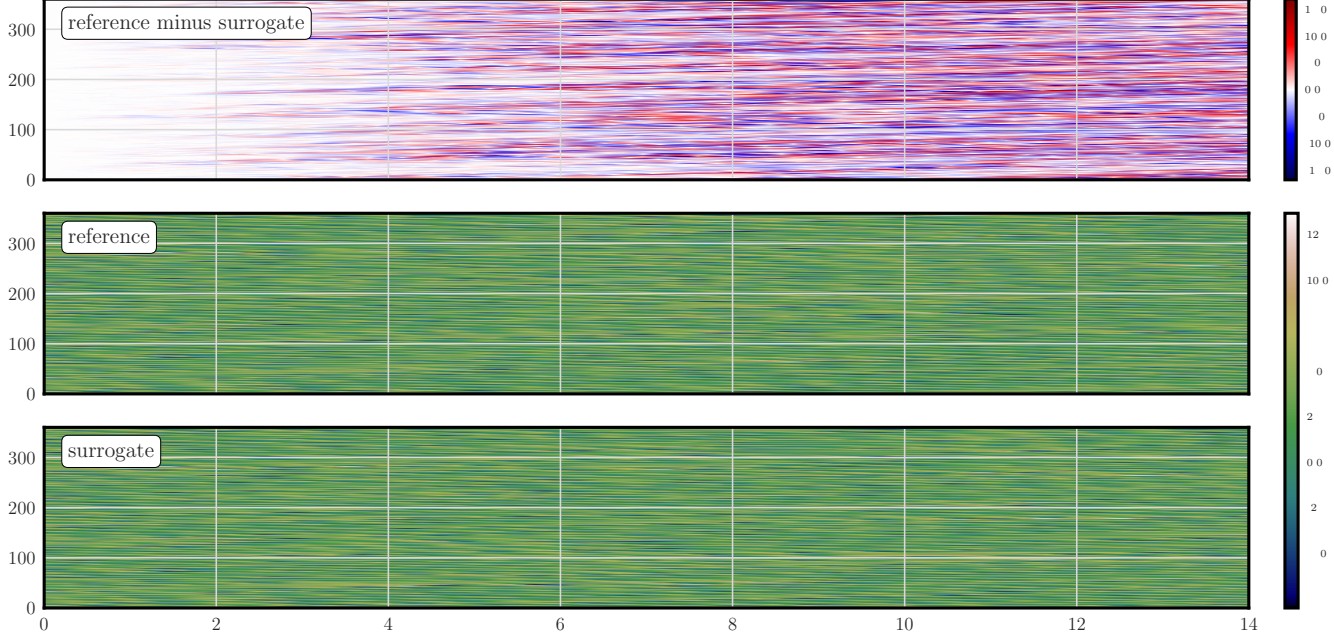

**Figure 9.** Density plot of the L05III reference and surrogate model trajectories, as well as their difference trajectory, as a function of the forecast lead time (in Lyapunov time unit).

function of the data assimilation problem is minimised using the adjoint of the surrogate resolvent which is explicitly derived. Analogies between the surrogate resolvent and a deep neural network have been discussed, as well as the impact of stability issues of the reference and surrogate dynamics.

The method has been applied to densely noiseless observed systems with identifiable reference model yielding a perfect

reconstruction close to machine precision (L63 and L96 models). It has also been applied to densely or partially observed, identifiable or non-identifiable model with or without noise in the observations (L96 and KS model). For moderate noise and sufficiently dense observation, the method is successful in the sense that the forecast is accurate beyond several Lyapunov times. The method has also been used as a way to infer a reduced model for a multi-scale observed system (L05-III model). The reduced model was successful in emulating slow dynamics of the reference model but could not properly account for the

impact of the fast unresolved scale dynamics on the slow ones. A subgrid parametrisation would be required or would have to be inferred.

Two potential obstacles have been left aside on purpose but should later be addressed. First, the model error statistics have not been estimated. This could be achieved using for instance an empirical Bayesian analysis based on a ensemble-based stochastic expectation maximisation technique. This is an especially interesting problem since the potential discrepancy

between the reference and the surrogate dynamics is in general non trivial. Second, we have used relatively short learning



time windows. Numerical efficient learning on longer windows will likely require using advanced weak constraint variational optimisation techniques.

In this paper, only autonomous dynamics have been considered. We could at least partially extent the method to non-autonomous systems by keeping a static part for the pure dynamics and consider time-dependent forcing fields. We have not numerically explored non-homogeneous dynamics but we have shown how to learn from them using non-homogeneous local representations.

A promising yet challenging path would be to consider implicit or semi-implicit scheme following for instance the idea in Chen et al. (2018). This idea is known in geophysical data assimilation as the continuous adjoint (see e.g., Bocquet, 2012). This would considerably strengthen the stability of the learning and forecast steps at the cost of more intricate mathematical developments.

If observations keep coming after the learning time window, then one can perform data assimilation using the ODE surrogate model of the reference model. This data assimilation scheme could only focus on state estimation, or it could continue to update the ODE surrogate model for the forecast.

*Data availability.* No data sets were used in this article

# Appendix A: Parametrisation of $\phi_{\mathbf{A}}$ for one-dimensional local and homogeneous representations

In this appendix, we show in the one-dimensional case how to parametrise $\phi_{\mathbf{A}}$ assuming locality and homogeneity of the representation. It is of the generic form:

$$[\mathbf{A} \bullet \mathbf{r}]_n = \sum_{p=0}^{N_p-1} A_{n,\pi(n,p)} r_p, \tag{A1}$$

where $\pi(n,p)$ is an integer such that $0 \leq \pi(n,p) < N_a$. We can treat the bias, linear and bilinear monomials separately into sectors, 0, 1 and 2, respectively. Set $0 \leq a_i \leq N_a - 1$ the indices which spans the columns of $\mathbf{A}$ for each of the three sectors $i$ and $0 \leq p_i \leq N_p - 1$ the indices which spans the entries of $\mathbf{r}$ for each of the three sectors $i$. Then, Eq. (A1) can be more explicitly written:

$$[\mathbf{A} \bullet \mathbf{r}]_n = A_{n,a_0(0)} r_{p_0(0)} + \sum_{l=0}^{2L} A_{n,a_1(n,l)} r_{p_1(n,l)}$$
$$+ \sum_{l=0}^{L} \sum_{m=0}^{2L-l} A_{n,a_2(n,l,m)} r_{p_2(n,l,m)}, \tag{A2}$$

where the dummy index $l$ for the linear terms browses the stencil, and the dummy indices $l, m$ for the bilinear monomials browse the stencil, in the same way as we did above to enumerate them. By enumeration, we find:

- For the bias sector, we have $p_0(0) = 0$ and $a_0(0) = 0$.





- For the linear sector, we have $a_1(n,l) = 1 + l$ and $p_1(n,l) = 1 + [n + l - L]$ , where $[n]$ means the index in $[0, N_x - 1]$ congruent to $n$ modulo $N_x$, in order to respect the periodicity of the domain.

- Finally, for the bilinear sector, we have $a_1(n,l) = 1 + 2L + 1 + \frac{1}{2}l(4L - l + 1) + m$ and $p_2(n,l,m) = 1 + N_x + [n - L + m](1 + L) + l$.

5  *Author contributions.*  MB first developed the theory, implemented, run and interpreted the numerical experiments, and wrote the original version of the manuscript. All authors have discussed the theory, the interpretation of the results and edited the manuscript. The four authors approved the manuscript for publication.

*Competing interests.*  The authors declare that they have no conflict of interest.

*Acknowledgements.*  MB is thankful to S. Ouala and R. Fablet for enlightening discussions. AC and JB have been funded by the project
10  REDDA (#250711) of the Norwegian Research Council. CEREA and LOCEAN are members of Institut Pierre–Simon Laplace (IPSL).



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
