# Peer review of "Data assimilation as a learning tool to infer ODE representations of dynamical models"

_Nonlinear Processes in Geophysics, 2019_

## Referee Comment (RC1) · Anonymous Referee #1 · 3 Apr 2019

General comments:

In this paper, authors want to learn the dynamics of a system from perfect or imperfect observations. To do so, they use the observations to build a surrogate model. Compared to recent papers that try to use Machine Learning (ML) algorithms to identify the surrogate model, authors in this paper are using Data Assimilation (DA) as an optimization tool. After identifying the dynamics with their technique, they apply the surrogate model to get forecasts and compare them to true simulation runs. Note that they use local representations to deal with high-dimensional models, an important challenge almost never treated in other papers using ML approaches.

The methodology is tested on different toy models, from the Lorenz attractor to the two-scale Lorenz-96 system. Results prove the reliability of the method, even if more

details about the identification part of the dynamics would be appreciated. Finally, authors pointed out interesting perspectives in the conclusion, especially the estimation of model error statistic for their surrogate model.

The paper is overall well written with a nice review of the existing and recent literature on data-driven approaches, including analogues, diffusion maps, reduced models and neural networks. Method is described with all the details. Experiments and results are well presented and discussed. However, I have some suggestions to improve the quality of the paper and avoid some misunderstandings. It concerns the title and some technical sections: these points are detailed below.

Problem of the title:

The authors should not use "deep learning" in their title. This is a confusing point because some readers might think that this paper is dealing with ML algorithms but this is not the case. However, the discussion in Sect. 3 about the link between the presented method (based on DA) and deep learning is interesting, especially when comparing equations (17) and (18). Moreover, the authors nicely show that using a weak constraint 4D-Var is a way of controlling the backpropagation (important part of deep learning methods).

Improve the readability and understanding of the method:

I think the authors should write more context in Sect. 2. First, in Sect. 2.1, the monomial basis should be discussed, giving more explanation and illustrating for instance with the Lorenz-63 (as Brunton did). I have the same remark in Sect. 2.2.1 where the drastic reduction of columns of A due to locality could be illustrated using for instance the L96 system. Then, Sect. 2.2.2 about the homogeneity (same behavior of the model at different locations) is hard to follow: again, authors should try to illustrate. Finally, entire Sect. 5 is very technical and I suggest to keep the important results/equations in the main text and move the rest to the appendix.

Other specific points:

- p3, l4-5 –> can you give a quick explanation of the difference with the neural network approach? - p6, l5 –> what do you mean by one-dimensional state space models? The models you introduce in Sect. 4 are all multi-dimensional, please clarify. - p16, l8-9 –> this is not always the case, especially when you use recurrent neural networks, that are not deep in practice. - p21, l16-17 –> in addition to Fig. 8, for different values of observation noise, I would like to have a look at the estimated coefficients of A along the DA cycles. - p23, l20 and Fig. 9 –> what do you mean by "long space-time stripes"? This point needs clarification with maybe a zoom on Fig. 9 to make this point clear.

Typo:

- p23, l2 –> "the the"

---

## Referee Comment (RC2) · Anonymous Referee #2 · 4 May 2019

This paper proposes to use data assimilation techniques to infer surrogate dynamical model from partial observations of reference model. Applications to several well-known toy model examples of geophysical flows are presented, where quality of the surrogate model is assessed by performing forecasts and comparing with those of the reference model. Obtained results demonstrate viability of the proposed methodology.

Overall, this is a solid paper that brings novel ideas from data-assimilation perspective to the burgeoning field of machine learning and data-driven modeling. Still some aspects would benefit from additional clarification, as suggested below.

1. p.6, lines 15-18. The locality assumption is obviously helpful since it reduces number of regressors, but I wouldn't go as far as claiming that long-range dependencies are precluded in geophysical applications. For example long-range teleconnections in

coupled ocean-atmosphere system are well established.

2. Section 2.2.2. The homogeneity assumption is also useful since it simplifies even more estimation of the surrogate model and obviously helps with robustness. On the other hand, the toy-model examples largely favor this assumption in a sense that the statistical properties of state variables are basically the same, namely x variables in L96, L05III and KS. However in real-world applications this is hardly the case, for example ENSO dynamics of sea surface temperatures in tropical Pacific which is very inhomogeneous. It would be helpful to have authors elaborate more on this point, i.e. in lines 23-25, p6.

3. Section 3.1. Having monomials in the surrogate model can bring numerical instabilities, so it is nice to see that authors have plans to deal with this issue – (Eq.43) and lines 10-14 on p.14. However it is not clear if this remedy has been applied for the examples presented.

4. The term "resolvent" was used multiple times, the meaning of it was not clear to me.

---

## Editor Comment (EC1) · Olivier Talagrand (Editor) · 7 May 2019

Following the reviews of the two referees, I first add an editing suggestion. In many places in Section 4 (for instance, p. 17, l. 14), the authors mention the number K of observation times in their numerical experiments. It would to useful to say to how many Lyapunov times this corresponds (the information is available in the paper for the reader to find out by himself, but an explicit mention would be useful).

At a more fundamental level, could it be possible to mention and discuss, if only briefly, the question of how to combine the approach proposed by the authors with an already known dynamical model ? That would naturally come through an a priori term – ln p(A) in Eq. (17). But how would p(A) be known ? And can we at this stage expect a

significant improvement of an existing model (used, for instance, for any kind of meteorological prediction) through the learning approach presented in the paper ?

---

## Author Comment (AC1) · 28 May 2019

May 28, 2019

**Subject:** Manuscript npg-2019-7: "Data assimilation as a deep learning tool to infer ODE representations of dynamical models", Response to Reviewer 1.

Dear Reviewer,

We wish to thank you for your comments and suggestions, that we have taken into

account to improve the manuscript. We respond below to these comments and tell how we modified the manuscript accordingly. A pdf file showing the differences between the original and the revised manuscript is provided.

*Problem of the title: The authors should not use "deep learning" in their title. This is a confusing point because some readers might think that this paper is dealing with ML algorithms but this is not the case. However, the discussion in Sect. 3 about the link between the presented method (based on DA) and deep learning is interesting, especially when comparing equations (17) and (18). Moreover, the authors nicely show that using a weak constraint 4D-Var is a way of controlling the backpropagation (important part of deep learning methods).*

We have removed "deep" from the title. As your subsequent comments suggest, the frontier between ML and data assimilation is not clear-cut. The method presented in this paper can certainly be seen as a learning method, while the optimisation of deep leaning networks is similar to that used in variational data assimilation.

*I think the authors should write more context in Sect. 2. First, in Sect. 2.1, the monomial basis should be discussed, giving more explanation and illustrating for instance with the Lorenz-63 (as Brunton did).*

As an example, we now list the monomials in the case of the L63 model.

*I have the same remark in Sect. 2.2.1 where the drastic reduction of columns of A due to locality could be illustrated using for instance the L96 system.*

As an example, we now list the monomials in the case of the L96 model for the case $L = 2$.

As for $\mathbf{A}$, we provide an example a few lines later in section 2.2.2.

Note also that we have improved Appendix A, which is closely related.

*Then, Sect. 2.2.2 about the homogeneity (same behavior of the model at different locations) is hard to follow: again, authors should try to illustrate.*

We have clarified the text and illustrated the enumeration of the coefficients of $\mathbf{A}$ in the L96 case, with $L = 2$, and assuming homogeneity in addition to locality.

*Finally, entire Sect. 5 is very technical and I suggest to keep the important results/equations in the main text and move the rest to the appendix.*

Following your suggestion, we have moved sections 2.5.1 and 2.5.2 to appendix B.

*p3, l4-5 → can you give a quick explanation of the difference with the neural network approach?*

We have slightly improved the sentence: "This marks a key distinction with respect to our approach where the dynamics to be determined are explicitly represented, as will be clarified later.", though we cannot go into much detail in the introduction.

*p6, l5 → what do you mean by one-dimensional state space models? The models you introduce in Sect. 4 are all multi-dimensional, please clarify.*

One-dimensional refers to the dimension of the physical space over which the model is defined (extend models), not the number of variables (resulting from the discretisation of the model). The sentence was indeed confusing. We have clarified this point in the revised manuscript.

*p16, l8-9 → this is not always the case, especially when you use recurrent neural*

*networks, that are not deep in practice.*

Thank you for spotting this issue. We have rephrased the sentence in order to avoid confusion: "Finally, we note that, as opposed to most practical deep learning strategies with a huge amount of weights to estimate, we have reduced the number of control variables (i.e. $\mathbf{A}$) as much as possible."

*p21, l16-17 → in addition to Fig. 8, for different values of observation noise, I would like to have a look at the estimated coefficients of A along the DA cycles.*

In the revised manuscript, we now plot the value of the coefficients of $\mathbf{A}$ as a function of the minimisation iteration index for the L96 model with and without noise in the observations.

*p23, l20 and Fig. 9 → what do you mean by "long space-time stripes"? This point needs clarification with maybe a zoom on Fig. 9 to make this point clear.*

We have clarified the sentence which becomes: "The emergence of error, i.e. the divergence from the reference, appears as long darker stripes on the density plot of the difference (close to zero difference values appear as white or light colour)."

The figure has been modified and a zoom over the difference density plot is now shown, where the stripes are patent.

*Typo: - p23, l2 → "the the"*

Corrected. Thank you.

Marc Bocquet, Julien Brajard, Alberto Carrassi, Laurent Bertino.

---

## Author Comment (AC2) · 28 May 2019

May 28, 2019

**Subject:** Manuscript npg-2019-7: "Data assimilation as a deep learning tool to infer ODE representations of dynamical models", Response to Reviewer 2.

Dear Reviewer,

We wish to thank you for your comments and suggestions, that we have taken into

account to improve the manuscript. We respond below to these comments and tell how we modified the manuscript accordingly. A pdf file showing the differences between the original and the revised manuscript is provided.

*1. p.6, lines 15-18. The locality assumption is obviously helpful since it reduces number of regressors, but I wouldn't go as far as claiming that long-range dependencies are precluded in geophysical applications. For example long-range teleconnections in coupled ocean-atmosphere system are well established.*

As a matter of fact, we were pointing to the absence of long-range instantaneous interactions which are indeed precluded in geophysical fluids. This does not prevent the building up of teleconnections and specific long-range dependencies. Following your comment, we have added the sentence: "This would not prevent potential specific long-distance dependencies (such as teleconnections)."

*2. Section 2.2.2. The homogeneity assumption is also useful since it simplifies even more estimation of the surrogate model and obviously helps with robustness. On the other hand, the toy-model examples largely favor this assumption in a sense that the statistical properties of state variables are basically the same, namely x variables in L96, L05III and KS. However in real-world applications this is hardly the case, for example ENSO dynamics of sea surface temperatures in tropical Pacific which is very inhomogeneous. It would be helpful to have authors elaborate more on this point, i.e. in lines 23-25, p6.*

Yes, we agree with you. We have further developed the paragraph by adding: "For realistic geofluids, the forcing fields (solar irradiance, bathymetry, boundary conditions, friction, etc.) are heterogeneous, so that the homogeneity assumption should be dropped. Nonetheless, the fluid dynamics part of the model would remain homogeneous. As a result, an hybrid approach could be enforced." Thank you for the suggestion.

*3. Section 3.1. Having monomials in the surrogate model can bring numerical instabilities, so it is nice to see that authors have plans to deal with this issue – (Eq.43) and lines 10-14 on p.14. However it is not clear if this remedy has been applied for the examples presented.*

Yes, these techniques are systematically enforced in the numerical experiments. They are especially useful for large windows. In practice, they only operate in the first few iterations of the minimisation. Later, a stable integration of the model is reached and the rectifier functions $\zeta$ act linearly. The text has been slightly improve to reflect this.

*4. The term "resolvent" was used multiple times, the meaning of it was not clear to me.*

The resolvent is the model integrated in time, say from $t_k$ to $t_{k+1}$, as opposed to the flow rate, which is instantaneous and not integrated. The term is now defined when first appearing.

Marc Bocquet, Julien Brajard, Alberto Carrassi, Laurent Bertino.

---

## Author Comment (AC3) · 28 May 2019

May 28, 2019

**Subject:** Manuscript npg-2019-7: "Data assimilation as a deep learning tool to infer ODE representations of dynamical models", Response to the Editor.

Dear Editor,
Dear Olivier,

[Figure]

We wish to thank you for your comments and suggestions, that we have taken into account to improve the manuscript. We respond below to these comments and tell how we modified the manuscript accordingly. A pdf file showing the differences between the original and the revised manuscript is provided.

*In many places in Section 4 (for instance, p. 17, l. 14), the authors mention the number $K$ of observation times in their numerical experiments. It would to useful to say to how many Lyapunov times this corresponds (the information is available in the paper for the reader to find out by himself, but an explicit mention would be useful).*

Following your suggestions, we have added the equivalent length in Lyapunov time units. Some of them are very short (below $1$) and correspond to noiseless observations experiments. Noisy observation experiments require to substantially increase $K$ as suggested in the original manuscript, which is further stressed in the revised manuscript. Thank you for the suggestion.

*At a more fundamental level, could it be possible to mention and discuss, if only briefly, the question of how to combine the approach proposed by the authors with an already known dynamical model ? That would naturally come through an a priori term $-\ln p(A)$ in Eq. (17). But how would $p(A)$ be known ? And can we at this stage expect a significant improvement of an existing model (used, for instance, for any kind of meteorological prediction) through the learning approach presented in the paper ?*

We have introduced a new short discussion (new subsection 3.1) in the beginning of the theoretical discussion part. Thank you for the suggestion.

Marc Bocquet, Julien Brajard, Alberto Carrassi, Laurent Bertino.
* * *
2019-7, 2019.

---

## Editor Decision (ED1)

(The page and line numbers are those of the version in which the modifications made by the authors are explicitly identified).

There is no significance in the order of the comments and suggestions below.

- P. 2, l. 20 … *renounce*  *physically-based* …

- P. 5, l. 10, … *3 variables, $x_0$, $x_1$, $x_2$* (not $x_3$)

- P. 7, ll. 15-16, … *hardly generate trivial flows.* → … *do produce non-uniform flows.* (I understand that is what you mean).

- P. 11, ll. 9 and 10, … *tangent linear* **operator** …

- P. 15, l. 10, … *with e.g., Fablet et al.* … Do you mean … *following e.g., Fablet et al.* … ?

- P. 16, l. 20, … *space-wise linear interpolation* … What is the observation operator $\mathbf{H}_k$ ? The remark is actually more general, and it might be useful to define more precisely, at this point and elsewhere, what $\mathbf{H}_k$ exactly is. My first understanding was that observations are made at every grid-point at each observation time (or on a 'dense' grid in the case when no grid is assumed to be *a priori* known). But that seems not to be the case here, and is obviously not the case for experiments described p. 23, ll. 1-6. Some clarification would be helpful.

- P. 10, l. 14, and p. 17, l. 8, $p(\mathbf{y}_O | \mathbf{A})$ (and not $p(\mathbf{y}_O, \mathbf{A})$)

- P. 18, l. 4, … *a lead time of 23*. I presume you mean … *a lead time of 23 Lyapunov times* (see also l. 12 further down) ?

- P. 25, l. 15 and p. 26, l.1, … *based on a ensemble-based stochastic* … I suggest … *built on an ensemble-based* …

- P. 11, ll. 14-15, change to … *the adjoint**s** $(\nabla_{xk} \mathbf{H}_k)^\mathrm{T}$ of the tangent […] operators are known, …*

---

## Author Response (AR2)

May 28, 2019

**Subject:** Manuscript npg-2019-7: "Data assimilation as a learning tool to infer ODE representations of dynamical models", Response to the Editor.

Dear Editor,
Dear Olivier,

Once again, thank you very much for your suggestions.

We have modified the text according to all of your suggestions with the exception of the remark on $p(\mathbf{y}_0, \mathbf{A})$ for which we believe our text was correct but confusing. We changed the corresponding sentences into "If, in addition, the dependence on $p(\mathbf{y}_0, \mathbf{A}) = p(\mathbf{y}_0|\mathbf{A})p(\mathbf{A})$ is neglected in..." and "...assuming $p(\mathbf{y}_0, \mathbf{A}) = p(\mathbf{y}_0|\mathbf{A})p(\mathbf{A})$ can be neglected...".

We have also made a few further English corrections to the text.

Best regards,

Marc Bocquet, Julien Brajard, Alberto Carrassi, Laurent Bertino.